# A Vaspin–HSPA1L complex protects proximal tubular cells from organelle stress in diabetic kidney disease

Atsuko Nakatsuka [1,2✉], Satoshi Yamaguchi[1], Jun Eguchi[1], Shigeru Kakuta[3], Yoichiro Iwakura [4], Hitoshi Sugiyama[5] & Jun Wada [1✉]

Proximal tubular cells (PTCs) are crucial for maintaining renal homeostasis, and tubular injuries contribute to progression of diabetic kidney disease (DKD). However, the roles of visceral adipose tissue-derived serine protease inhibitor (vaspin) in the development of DKD is not known. We found vaspin maintains PTCs through ameliorating ER stress, autophagy impairment, and lysosome dysfunction in DKD. Vaspin−/− obese mice showed enlarged and leaky lysosomes in PTCs associated with increased apoptosis, and these abnormalities were also observed in the patients with DKD. During internalization into PTCs, vaspin formed a complex with heat shock protein family A (Hsp70) member 1 like (HSPA1L) as well as 78 kDa glucose-regulated protein (GRP78). Both vaspin-partners bind to clathrin heavy chain and involve in the endocytosis. Notably, albumin-overload enhanced extracellular release of HSPA1L and overexpression of HSPA1L dissolved organelle stresses, especially autophagy impairment. Thus, vapsin/HSPA1L-mediated pathways play critical roles in maintaining organellar function of PTCs in DKD.

[1] Department of Nephrology, Rheumatology, Endocrinology and Metabolism, Okayama University Graduate School of Medicine, Dentistry and Pharmaceutical Sciences, Kita-ku, Okayama 700-8558, Japan. [2] Division of Kidney, Diabetes and Endocrine Diseases, Okayama University Hospital, Kita-ku, Okayama 700-8558, Japan. [3] Department of Biomedical Science, Graduate School of Agricultural and Life Sciences, the University of Tokyo, Tokyo 113-8657, Japan. [4] Research Institute for Biomedical Sciences, Tokyo University of Science, Chiba 278-0022, Japan. [5] Department of Human Resource Development of Dialysis Therapy for Kidney Disease, Okayama University Graduate School of Medicine, Dentistry and Pharmaceutical Sciences, Kita-ku, Okayama 700-8558, Japan. ✉email: atsuko-n@md.okayama-u.ac.jp; junwada@okayama-u.ac.jp

Diabetic kidney disease (DKD) has been a leading cause of end-stage kidney disease. Recently, obesity is recognized as a major risk factor for a sustained decline of renal function in DKD patients. Therefore, obesity-related metabolic disturbances are emerging contributors for renal impairment besides hyperglycemia. Despite a high impact of tubulointerstitial damage on renal outcome[1], the glomerular legions, such as hypertrophy and sclerosis, attracted the main interests of researchers and a pathophysiological role of tubular injuries during the progression of DKD with obesity is not fully explored.

Proximal tubular cells (PTCs) provide several physiological functions for the reabsorption and transport of glucose, amino acids, electrolytes, and solute, that are major tasks for these specialized cells. The reabsorbed macromolecules including albumin are degraded, and then recycled or secreted[2]. PTCs also take part in renal gluconeogenesis[3], and vitamin D synthesis[2]. Therefore, high energy demand of PTCs is associated with a large number of mitochondria[1]. The diverse functions of PTCs require the capabilities of each organelle which is integral element to maintain cellular viability. In obesity and diabetes, several cellular stresses, such as endoplasmic reticulum (ER) stress, oxidative stress and lipotoxicity are enhanced in systemic tissues[4–7]. Autophagy impairment is also observed in obesity[8,9]. Over-influx of nutrients and macromolecules into PTCs is generally harmful to the cells, however, the regulatory mechanisms and participating molecules to prevent organelles dysfunction under metabolic stresses have not been identified.

We previously identified an adipokine, visceral adipose tissue-derived serine protease inhibitor (vaspin)[10], and reported the beneficial effects of vaspin on obesity and type 2 diabetes. In brief, body weight gain, fatty liver and insulin resistance were improved in transgenic (Tg) mice under high fat-high sucrose (HFHS) diet, while knockout (KO) mice revealed deterioration of hepatosteatosis and insulin resistance[11,12]. Vaspin gene transfer ameliorated balloon-injured intimal proliferation in WKY rats and it protected from apoptosis of cultured human aortic endothelial cells[11,12]. Vaspin contributes to favorable roles on hepatocyte or endothelial cell via binding to cell surface 78 kDa glucose-regulated protein (GRP78)[11,12]. Although GRP78 is known as an ER stress-associated protein, recently it is reported to translocate from ER to plasma membrane by several cellular stresses[13]. In hepatocyte, vaspin binds to cell surface GRP78/murine tumor cell DnaJ-like protein 1 and transduces intracellular signaling, such as phosphorylation of serine/threonine protein kinase (Akt) and AMP-activated protein kinase[11]. In endothelial cells, kringle 5 binds to cell surface voltage dependent anion channel (VDAC) and increases intracellular $Ca^{2+}$, subsequently induces apoptosis. Vaspin binds to cell surface GRP78/VDAC complex and competitively inhibits kringle 5-induced apoptosis[12].

Since the adipokines, such as leptin, adiponectin, resistin and visfatin, affect renal homeostasis and progression of chronic kidney disease in obesity[14], it led us to investigate a role of vaspin in the obesity-related kidney disease. In vaspin−/− mice under HFHS diet, prominent vacuoles were observed in PTCs, and they were turned out to be enlarged lysosome. Recently it is reported that high fat diet-induced lysosomal dysfunction and impaired autophagic flux contribute to lipotoxicity in the kidney and phospholipid accumulation was enhanced in enlarged lysosomes[15]. Kuwahara et al. reported that fatty acid rich albumin, which is reabsorbed into proximal tubular cells through megalin, causes autophagy impairment with large autolysosome formation[16]. Lysosome, that contains numerous proteases and digests waste materials, is called suicide bag and a leakage of protease into cytosol induces cell death. Here, we show protective roles of vaspin against injuries in PTCs through ameliorating enhanced ER stress, autophagy impairment, lysosomal membrane

permeabilization (LMP) and subsequent cell death. We also revealed the beneficial mechanisms of vaspin on PTCs through vaspin interactive molecules, i.e., GRP78 and heat shock protein family A (Hsp70) member 1 like (HSPA1L); both molecules belong to heat shock protein 70s family. GRP78 is well known for the localization on endoplasmic reticulum (ER) and repairing misfolded proteins[17]. Cell surface GRP78 as a receptor and its distinct role beyond the ER chaperone have been reported[13]. We identified a unique role of GRP78 in endocytosis process of vaspin into PTCs. We also identified vaspin-interacting molecule, HSPA1L, has 91% homology to HSPA1A; the major stress-inducible members of the Hsp70s, however, its function and regulation are not well investigated[18]. We first demonstrated HSPA1L and vaspin complex was internalized by endocytosis process mediated by clathrin heavy chain and ameliorated the obesity-induced LMP, inflammasome activation, and autophagy failure. Furthermore, we revealed that an overload and uptake of albumin into PTCs increased the secretion of HSPA1L into culture medium and resulted in reduced levels of intracellular HSPA1L, suggesting an interest mechanism albuminuria-induced disruption of obesity-induced organellar dysfunction in PTCs. The current study presents a crucial role for HSPA1L in endocytosis and beneficial function of vaspin, and suggests a potential of vaspin in a treatment of PTC injuries under DKD.

## Results

**Prominent vacuolation of proximal tubular cells was observed in vaspin−/− mice fed with HFHS diet.** Vaspin transgenic (Tg) mice, wild type mice and vaspin−/− mice were fed with STD and HFHS chow and PAS staining of kidney tissues at 30 weeks of age are shown in Fig. 1a–f. In vaspin−/− mice fed with HFHS diet, prominent vacuolations were observed in renal tubular cells (Fig. 1c). These vacuoles were positive for lysosomal membrane marker, lysosomal-associated membrane protein 1 (lamp1) (Fig. 1g–o), PTCs marker, aquaporin 1 (AQP1) (Supplementary Fig. S1a, b), and toluidine blue (Supplementary Fig. S1b). In electron microscopy (EM), enlarged lysosomes filled with electron-dense materials were observed (Supplementary Fig. 1c). These findings indicated that lysosomal enlargement in proximal tubular cells was enhanced in diet-induced obese vaspin−/− mice, while vacuolar formation was ameliorated in vaspin Tg mice. In Vaspin Tg, wild type (WT) and Vaspin−/− mice under STD, the vacuolations in tubules were absent, while HFHS diet significantly increased the vacuolation score. Comparison among Vaspin Tg, WT, Vaspin−/− mice under HFHS diet demonstrated that the score of Vaspin Tg mice was significantly lower than WT and Vapsin−/− mice (Supplementary Fig. 1d). There were no statistical differences of dilatation score among Vaspin Tg, WT and Vaspin−/− mice under STD or HFHS diet (Supplementary Fig. 1e). In immunohistochemical staining of tubules, HFHS diet-increased p62 accumulation was enhanced in Vapsin−/− mice and it was ameliorated in Vaspin Tg mice (Supplementary Fig. S1f, g). As regards the glomerular legion, HFHS diet-induced increase of mesangial matrix index was inhibited in Vaspin Tg mice compared with WT and Vapsin−/− mice (Supplementary Fig. 1h and 1i). Urinary albumin tended to increase in the mice fed with HFHS diet compared with STD, although there were no statistical differences. Under HFHS diet, there were no differences of urinary albumin among Vaspin Tg, WT and Vapsin−/− mice (Supplementary Fig. S1j).

**Vaspin ameliorated saturated fatty acid- and excessive ER stress-induced lysosome enlargement, p62 accumulation and apoptosis in HK2 cells.** A mouse model fed with HFHS chow, higher calorie diet with fatty acid and sucrose, represents features

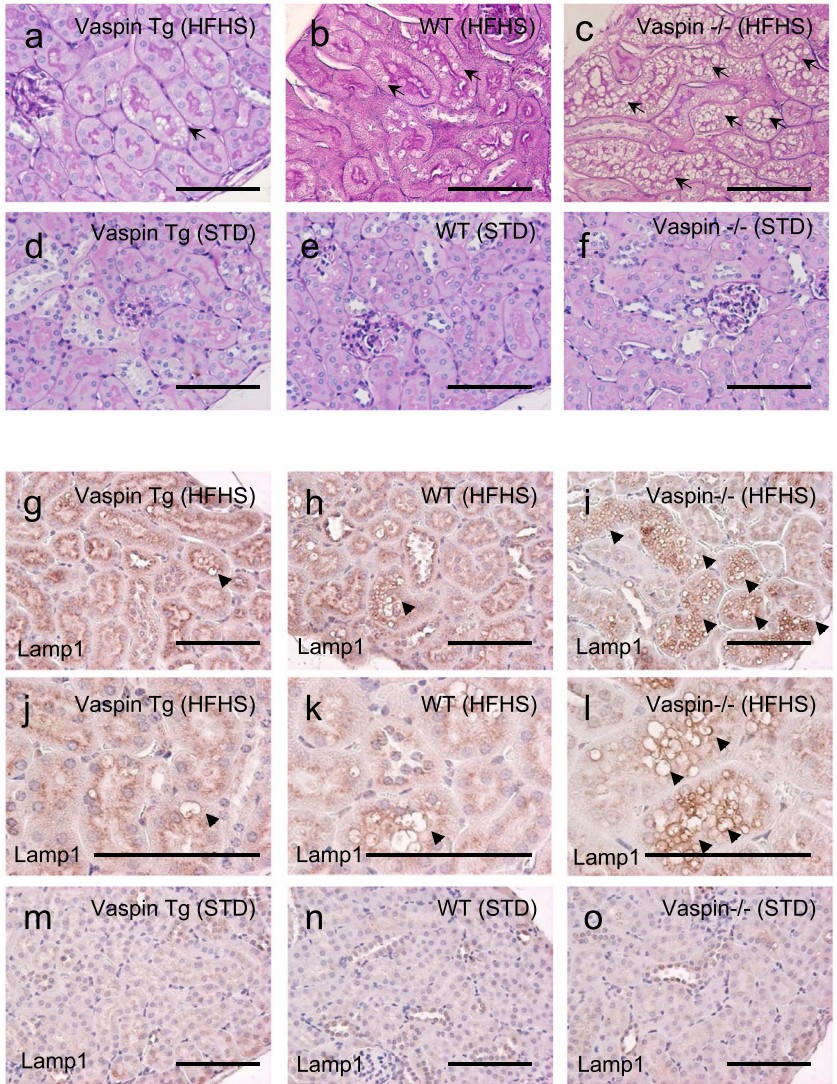

**Fig. 1 Light microscopic photographs of Vaspin Transgenic (Tg), wild type (WT), Vaspin−/− mice under high fat-high sucrose (HFHS) diet, or standard diet (STD) at 30 weeks of age.** Periodic Acid-Schiff staining of kidney tissues (**a–f**). In Vaspin−/− mice fed with HFHS diet, prominent vacuolations are observed in renal tubular cells (**c**), while they are ameliorated in Vaspin Tg mice (**a**). Vacuolations are shown by arrows. Immunohistological staining of lysosomal-associated membrane protein 1 (Lamp1) (**g–o**). These vacuoles are positive for lysosomal-associated membrane protein 1 (Lamp1). Lamp1-positive lysosomes are shown by arrow heads. In Vaspin−/− mice under HFHS diet, the number of large lysosomes is increased (**i, l**) compared with WT mice (**h, k**). HFHS diet-induced lysosomal enlargement is inhibited in Vaspin Tg mice (**g, j**). Different magnifications are used for images **g–i** and **j–l**, Bar = 100 μm. Immunoperoxidase staining was repeated twice.

with metabolic syndrome. It is well known that saturated fatty acid induces lipotoxicity in multiple type of organs[19]. We focused on palmitate-induced lipotoxicity in lysosomes of PTCs. In HK2 cells, lysosome was stained (Fig. 2a) and lysosomal membrane marker lamp1 (Fig. 2b). Palmitate administration promoted lysosome enlargement. The expressions of phosphorylated eIF2α, GRP78 and C/EBP homologous protein (CHOP) were increased by palmitate administration and inhibited by recombinant human vaspin protein (rhVaspin) (Fig. 2c, d). Palmitate-induced TUNEL-positive apoptotic cell death also suppressed by rhVaspin (Fig. 2e). In addition, rhVaspin suppressed palmitate-induced accumulation of p62 with statistical significance, but not LC3 (Fig. 2c, f). The p62 protein, which is ubiquitin-binding scaffold protein, and LC3 cooperates to deliver the waste materials to autophagosomes for degradation. In general, an increase in both LC3-II and p62 parallels in the enhancement of autophagy or lysosomal

dysfunction. Since we observed that rhVaspin reversed p62 but not LC3, we further moved to investigate the effects of rhVaspin on established ER stress inducers.

In metabolic syndrome, ER stress responses are enhanced in various tissues, such as adipose tissue, liver, pancreas and kidney[20]. ER stressors, tunicamycin and thapsigargin, increased the number of enlarged lysosome and they were reduced by rhVaspin in PTCs (Supplementary Fig. S2a, c). These ER stressors increased expressions of GRP78, phosphorylated eIF2α, CHOP as is the case with palmitate administration, and rhVaspin clearly suppressed the expressions of these ER stress associated molecules such as GRP78, phosphorylated eIF2α and CHOP (Supplementary Fig. S2b, d). An accumulation of p62, which was induced by tunicamycin and thapsigargin, was also inhibited by rhVaspin (Supplementary Fig. S2b, d). In TUNEL assay, rhVaspin suppressed ER stressor-induced apoptosis (Supplementary Fig. S2e, f).

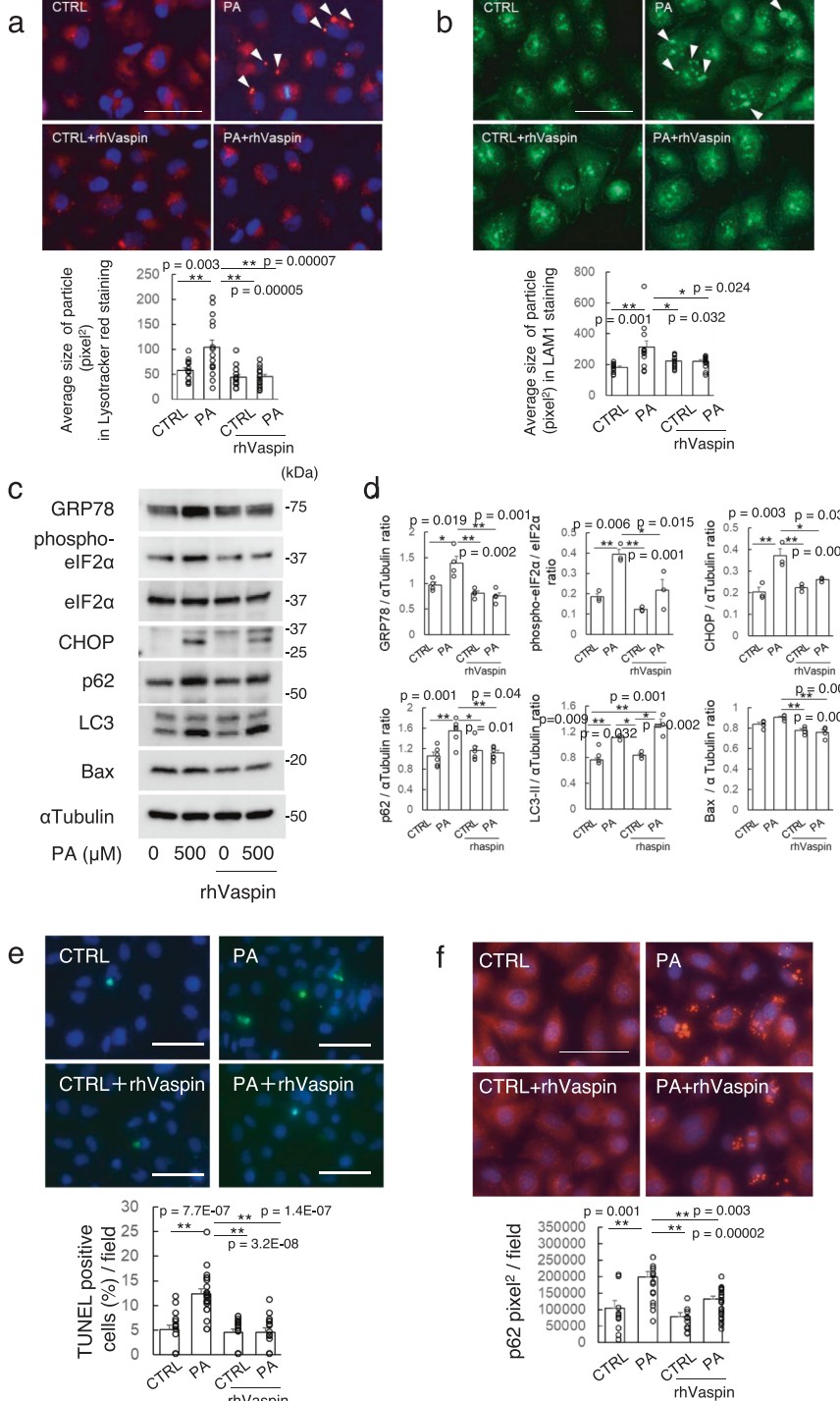

**Fig. 2 Palmitate (PA)-induced lysosomal enlargement along with ER stress enhancement in HK2 cells.** HK2 cells were cultured with 500 μM palmitate (PA) or control medium (CTRL) for 24 h. Recombinant human vaspin (rhVaspin) was administrated at 100 ng/ml with PA or CTRL for 24 h. Lysosome were stained by Lysotracker (red) (**a**) and Lamp1 (green) (**b**). Bar = 50 μm. Average size of Lysotracker Red-positive particle (**a**) and LAMP1-positive particles (**b**) are analyzed. PA-induced p62 accumulation and lysosomal enlargement are inhibited by rhVaspin. Bonferroni test is used for assessment. **c** Western blot analyses. **d** Densitometric analysis of western blots, $n = 3$–4 independent experiments. One-way ANOVA with Tukey–Kramer method is used for assessment, except for CHOP which is analyzed by Bonferroni test. **e** TUNEL assay of HK2 cells cultured with 0 or 250 μM PA for 24 h. Bar = 100 μm. Lower panel shows percentage of HK2 cells with TUNEL positive apoptotic cells per field. $N = 16$–21 independent cells. One-way ANOVA with Tukey–Kramer method is used. **f**. Cells were fixed with methanol, then stained with anti-p62 antibody. Bar = 50 μm. Lower panel statistical analysis of pixel$^2$ of p62 area, $n = 10$–25 independent cells. One-way ANOVA with Tukey–Kramer method is used. All data are presented as mean ± S.E., *$p < 0.05$, **$p < 0.01$. Western blots and fluorescence studies were repeated twice.

**Lysosomal membrane permeabilization (LMP), induced by palmitate and ER stressor, promoted NLRP3 inflammasome activation**. In aged tissues, it is reported that large lysosomes appear as a result of decreasing degradation capacity together with increasing undigested materials[21]. Large lysosome was also observed in vaspin−/− mice with HFHS diet-induced obesity. Large lysosome is vulnerable and susceptible to LMP[17,22]. Therefore, we investigated the LMP of PTCs and relocalization of cathepsin B to cytosol, which is major protease included in lysosome. In immunofluorescence staining of unstressed cells, cathepsin B of lysosome is shown as dot signals, while in stressed cells, cathepsin B leaking into cytosol is visible as diffuse signals[23]. In palmitate treated HK2 cells, diffused cathepsin B signals were observed, while rhVaspin administration maintained cathepsin B as dot signals (Fig. 3a). Next, we performed Western blot analysis of cathepsin B using cytosol-fraction lysates of HK2 cells, and palmitate-induced shift of cathepsin B into cytosol was inhibited by rhVaspin (Fig. 3b, c). ER stressors, tunicamycin and thapsigargin, also promoted leakage of cathepsin B into cytosol in immunofluorescence (Fig. 3d). In Western blot analysis, tunicamycin and thapsigargin increased cytosolic cathepsin B and the leakage was suppressed by rhVaspin, although there were no statistical differences (Fig. 3e, f). Recently, it is reported that galectin-3 puncta assay is useful to screening and visualization of LMP with high sensitivity[24]. In galectin-3 assay, we observed that tunicamycin and thapsigargin increased a number of galectin-3 dots co-localized with lamp1 similar to palmitate treatment, and rhVaspin inhibited ER stressor- and palmitate-induced galectin-3 accumulation into lysosome (Supplementary Figs. S3 and S4). Aits et al. reported that thapsigargin did not induce LMP[24], while Shin et al. demonstrated that an induction of protein misfolding by puromycin, thapsigargin, or geldanamycin resulted in inflammasome activation though lysosomal cathepsin B release[25]. Although it is controversial whether thapsigargin promote LMP, in our settings, palmitate and ER stressors induced LMP with leakage of cathepsin B into cytosol and rhVaspin inhibited LMP with release of cathepsin B into cytosol.

Recently, it is known that innate immune system-driven inflammatory processes, such as Toll-like receptors signaling and NLR (nucleotide-binding oligomerization domain-like receptor) family, pyrin domain-containing 3 (NLRP3) inflammasome pathways, can lead to apoptosis, tissue fibrosis, and organ dysfunction[26]. There are various NLRP3 inflammasome activators, such as microorganisms (pathogen-associated molecular patterns; PAMPs), endogenous danger signals (damage-associated molecular patterns; DAMPs) and environmental irritants[27]. Cathepsin B is also one of NLRP3 inflammasome activators, because cathepsin B inhibitor suppressed artificial lysosome disruption-induced NLRP3 inflammasome activation[27]. In our experiment, tunicamycin and thapsigargin increased expressions of NLRP3, cleaved caspase 1 and IL1-β, and their expressions were inhibited by rhVaspin (Fig. 4a–d). Subsequently, TUNEL positive apoptotic cells in tubules were increased in vaspin−/− mice under HFHS diet, while HFHS induced apoptotic cells was suppressed in vaspin Tg mice (Fig. 4e, f).

In summary, vaspin is a molecule that inhibits LMP, NLRP3 inflammasome activation and protect PTCs from apoptosis under metabolic stresses.

**Cell surface GRP78, a partner of vaspin, in proximal tubular cells is increased by ER stressor or palmitate**. We previously reported that vaspin forms a complex with GRP78 in hepatocytes and vascular endothelial cells[11,12]. We generated plasmid expressing 3xFLAG-Vaspin using p3xFLAG CMV expression vector and plasmids were transfected into HK2 cells. The complex formation of vaspin with GRP78 was observed in PTCs line (HK2 cells), in which p3xFLAG-Vaspin was overexpressed and ER stress was induced by tunicamycin or thapsigargin (Fig. 5a). To demonstrate the protein interactions of vaspin with GRP78 and HSPA1L, HEK293T cells were employed because of better scale-up efficiency for protein preparation in immunoprecipitation studies. As a result, in HEK293T cells overexpressing p3xFLAG-Vaspin and pGFP-GRP78, we confirmed a binding of vaspin and GRP78 (Fig. 5b).

GRP78 is reported to localize on cell-surface as a receptor and transduce survival signals into cancer cells[13]. To demonstrate the specific cellular responses of PTCs to various stress, we used HK2 cells in following experiments. We found that cell-surface GRP78 was prominently increased by tunicamycin, thapsigargin and palmitate administration, while total cell surface protein loadings were equal (Fig. 5c). Vaspin levels of whole cell lysate were not altered in HK2 cells treated with ER stressor or palmitate (Fig. 5d), despite of increase in cell surface GRP78 (Fig. 5c).

In GRP78 depleted cells, a beneficial effect of rhVaspin on HK2 cells, ameliorating accumulation of p62 caused by ER stress enhancer, disappeared (Supplementary Fig. S5). ER stressors-induced upregulation of phosphorylated eIF2α, ATF4 and CHOP were suppressed by rhVaspin administration in control HK2 cells, while the beneficial effects were not observed in GRP78 knockdown cells (Supplementary Fig. S5). Taken together, vaspin coordinated with GRP78 and demonstrated protective effects, such as reducing ER stress enhancement and autophagy impairment on PTCs.

**HSPA1L is identified as vaspin interactive molecule in PTCs**. p3xFLAG-Vaspin plasmids were transfected into HK2 cells and soluble proteins were purified by an ANTI-FLAG M2 affinity agarose gel and subjected to SDS-PAGE and silver staining (Fig. 6a). Visible bands were excised, in-gel digested with trypsin, and analyzed by liquid chromatography-tandem mass spectrometry. As a result, several candidates were identified as vaspin-interactive molecules (Supplementary Table S2). Then, we confirmed a binding of vaspin to HSPA1L by immunoprecipitation and Western blot analysis (Fig. 6b–d). The interaction of vaspin and HSPA1L was also confirmed in vivo. By using vaspin antibody, we immunoprecipitated the kidney lysates derived from the mice injected with rhVaspin, then we subjected them to SDS-PAGE and the detection with HSPA1L antibody (Supplementary Fig. S6a). HSPA1L belongs to Hsp70 family, and shares 91% homology with HSPA1A[18]. Previous report demonstrated that HSPA1L is mainly expressed in testis, and located in cytosol or nucleus[18]. Recently, Wang et al. reported mRNA expressions of Hspa1l in mouse heart, liver, spleen, ling, kidney, brain, muscle, and testis[28]. We also detected the gene expression of Hspa1l in C57BL/6J mouse kidney tissues (Supplementary Fig. S6b). We explored HSPA1L protein expressions in major organs of C57BL/6J mice and detected in various organs including kidney (Fig. 6e and Supplmentary Fig. S6d). Since albuminuria is major driving force for the progression of renal dysfunction and BSA is known to induce the lysosomal dysfunction and accumulation of p62[15,29], we further applied BSA in HK2 cell culture. In in vitro analysis of HK2 cells cultured with 25 mM high glucose, 100 nM insulin, 10 mg/ml BSA, 1 μg/ml tunicamycin, and 500 μM palmitate, HSPA1L expression was significantly decreased by BSA administration in HK2 cell culture (Fig. 6f, g). Mannitol was used as osmotic control of BSA, and it did not alter the expression of HSPA1L (Supplementary Fig. S6c).

BSA-induced decrease of HSPA1L was accompanied with p62 accumulation, reflecting impairment of autophagosomal degradation, and an overexpression of HSPA1L suppressed BSA-induced

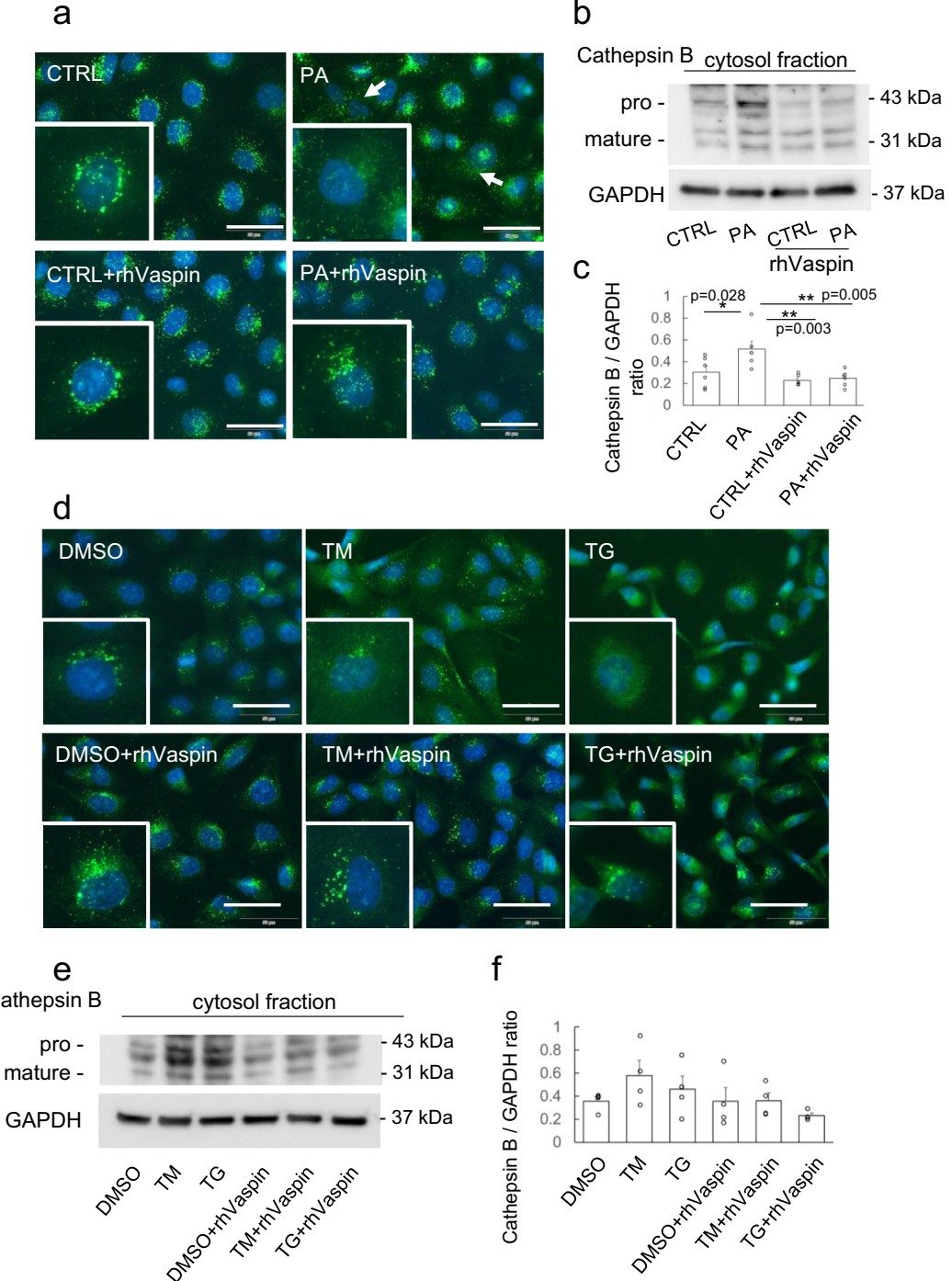

**Fig. 3 Cathepsin B leakage into cytosol induced by palmitate-mediated ER stress. a–c** HK2 cells were cultured with 500 μM palmitate (PA) or 0 μM (CTRL) for 24 h. **d–f** HK2 cells were treated with DMSO or 1 μg/ml tunicamycin (TM) or 1 μM thapsigargin (TG) for 24 h. rhVaspin was administrated at 100 ng/ml for 24 h. The cells were fixed, permeabilized and stained with anti-cathepsin B antibody (green) with DAPI (blue) (**a**, **d**). Bar = 50 μm. The insets show high-power fields. Cytosol fraction proteins were isolated and subjected into Western blot analyses of Cathepsin B and GAPDH (**b** and **e**). Sum of pro- and mature Cathepsin B levels in cytosolic protein are analyzed with one-way ANOVA followed by Tukey–Kramer method (**c** and **f**). Data are presented as mean ± S.E., $n = 4$–6 independent experiments, *$p < 0.05$. Western blots and immunofluorescence studies were repeated twice.

p62 accumulation (Fig. 6h and Supplementary Fig. S6e), indicating that HSPA1L promoted autophagy. rhVaspin administration also inhibited BSA-induced p62 accumulation (Fig. 6i and Supplementary Fig. S6e), therefore vaspin has a role in regulating autophagy through binding with HSPA1L. HSPA1L is highly homologous to

hsc70 (*HSPA8*), which is well known as a player of chaperon mediated autophagy and binds to lamp2 on the lysosomal membrane[30]. Thus, it is plausible that HSPA1L also promotes chaperon mediated autophagy. We further investigated whether HSPA1L binds to lamp2 as well as hsc70 and ascertained a binding

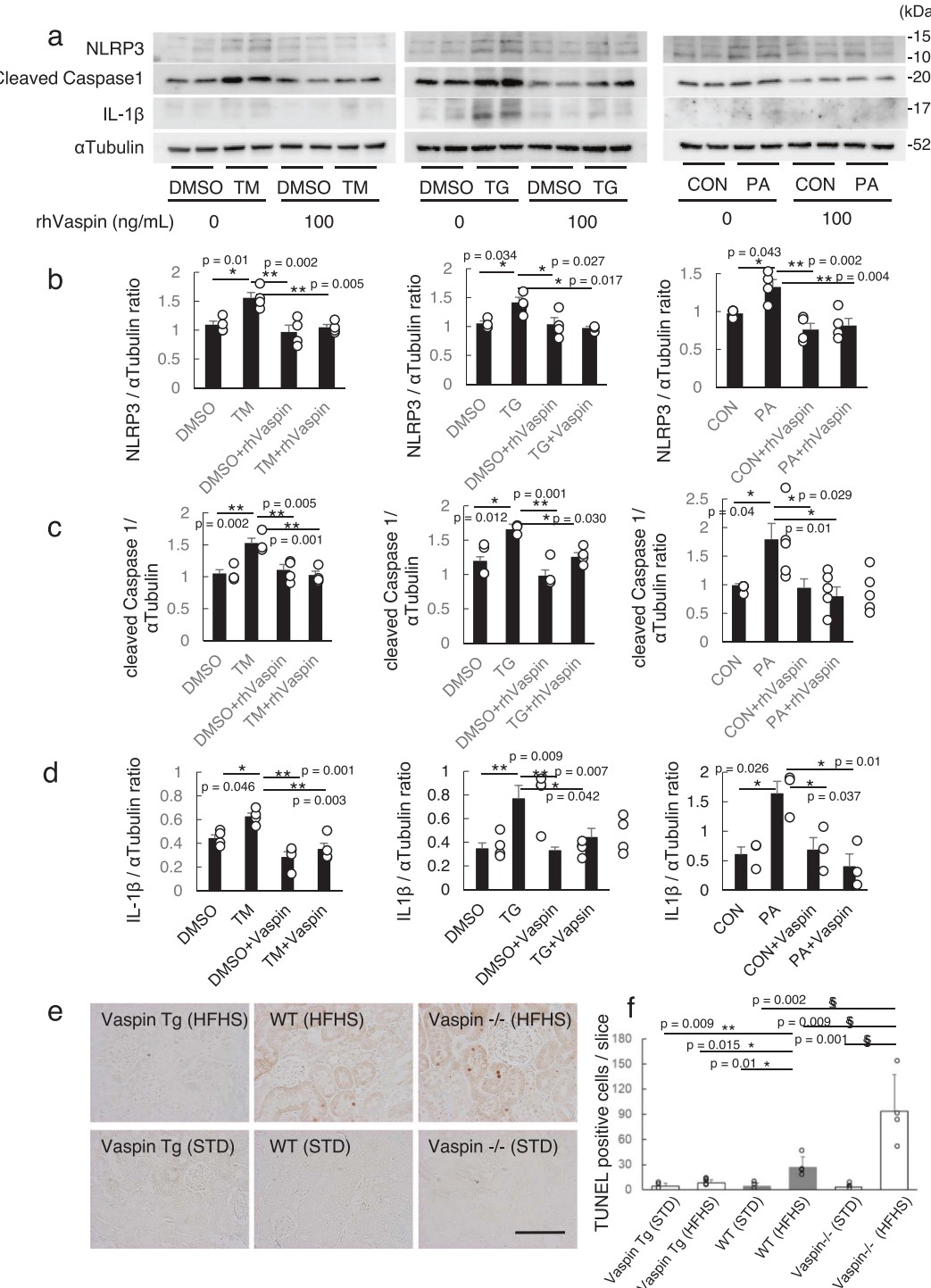

**Fig. 4 Inhibition of NLRP3 inflammasome activation by recombinant human vaspin (rhVaspin).** HK2 cells were cultured with DMSO or 1 μg/ml tunicamycin (TM) or 1 μM thapsigargin (TG) or 500 μM Palmitate (PA) for 24 h. rhVaspin was co-administrated at 100 ng/ml for 24 h. **a**. Western blot analysis of NLRP3 infammasome pathway related proteins; NLRP3, Caspase 1 and IL-1β. **b–d**. Densinometric quantification of Western blots. NLRP3 (**b**), cleaved Caspase 1(**c**), and IL-1β (**d**) are normalized by αTubulin. Data are shown as means ± S.E., n = 4–5 independent experiments, *p < 0.05, **p < 0.01. **e**. TUNEL staining. Bar = 100 μm. **f** Number of TUNEL-positive tubular cells per one slice are counted, then analyzed with one-way ANOVA followed by Tukey-Kramer method, n = 3–5 independent mice. Data are presented as mean ± S.D. *p < 0.05, **p < 0.01 in comparison of Vaspin Tg and WT mice under HFHS or STD. *§ <;0.05, **§§ < 0.01 in comparison of Vaspin−/− and WT mice under HFHS or STD. Western blots were repeated twice.

of HSPA1L to lamp2 (Supplementary Fig. S7). Interestingly, in cultured hepatic cell line, H4-II-E-C3 cells, there were no changes of HSPA1L and p62 levels by BSA administration (Fig. 6j and Supplementary Fig. S6e). BSA-induced depletion of HSPA1L may be one of characteristic findings of renal PTCs.

**BSA-induced depletion of HSPA1L in PTCs is associated with increased HSPA1L secretion.** To investigate the mechanism of a BSA-induced decrease of HSPA1L protein levels in PTCs, we first explored whether a degradation of HSPA1L was enhanced by ubiquitin-proteasome pathway. Although an administration

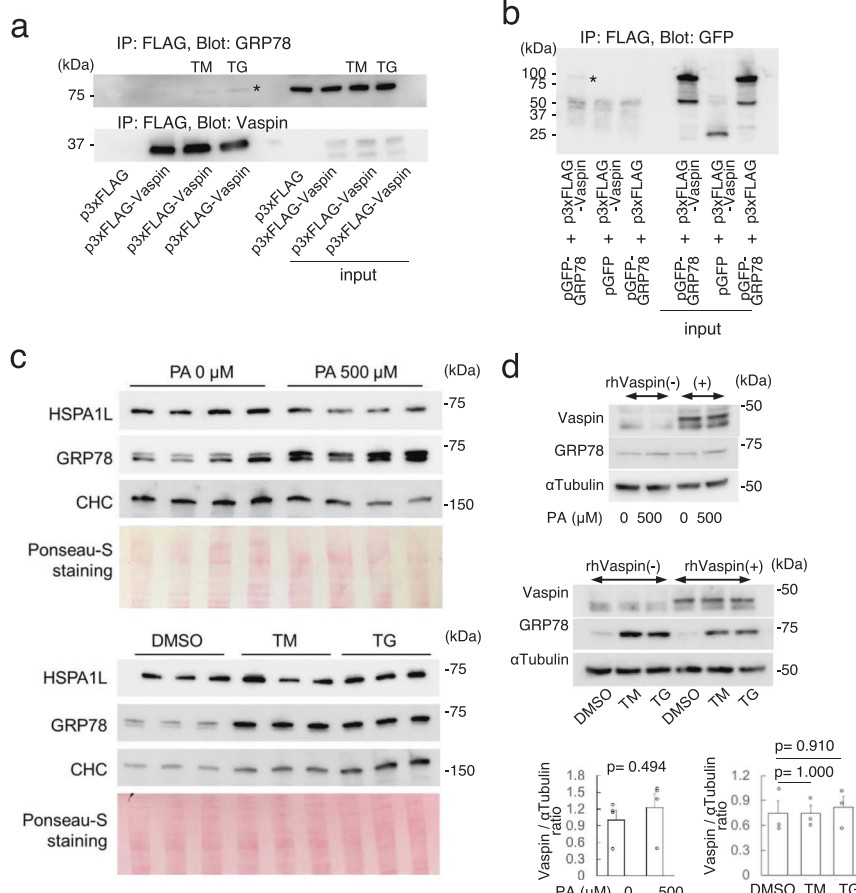

**Fig. 5 Localization of GRP78 on cell surface and formation of a complex with Vaspin. a** Immunoprecipitation using HK2 cell expressing p3xFLAG-Vaspin, with ANTI-FLAG M2 affinity agarose gel. Under 1 μg/ml tunicamycin (TM) or 1 μM thapsigargin (TG) induced ER stress enhanced condition, a binding of vaspin and GRP78 was visualized (asterisk). **b** Immunoprecipitation using HEK293T cells expressed p3xFLAG-Vaspin and pGFP-GRP78 with ANTI-FLAG M2 affinity agarose gel. GFP-GRP78 interacts with 3xFLAG-Vaspin (asterisk). **c** Western blot analyses of GRP78, HSPA1L and clathrin heavy chain (CHC) using cell surface protein of HK2 cells cultured with 500 μM palmitate (PA), 1 μg/ml TM and 1 μM TG. Ponceau-S staining shows equal amount of protein loading. Cell surface GRP78 is prominently increased by PA, TM and TG. In contrast, HSPA1L is mildly increased by TM and TG, but rather decreased by PA. **d** Western blot analyses of HK2 cells cultured with 500 μM PA, 1 μg/ml TM and 1 μM TG with or without 100 ng/ml rhVaspin for 24 h. Lower panel shows statistical analysis of Vaspin protein levels of HK2 cells under PA or TM or TG, which are normalized by αTubulin. $N = 3$–4 independent experiments. Unpaired 2-tailed by Student's $t$ for 2 groups comparison or one-way ANOVA with Tukey-Kramer method for multiple comparison are used. Western blots were repeated twice.

of proteasome inhibitor MG132 increased HSPA1L protein levels, an BSA-induced decrease of HSPA1L was not fully recovered (Fig. 7a). Next, we examined whether extracellular secretion of HSPA1L is promoted by BSA administration and resulted in subsequent decrease in intracellular HSPA1L protein. HSPA1L in culture medium, which was secreted from HEK293T cells overexpressing p3xFLAG-HSPA1L, was increased by BSA administration (Fig. 7b). These findings demonstrate that an intracellular HSPA1L level is regulated by exocytosis and proteasomal degradation. In addition, rhVaspin administration suppressed BSA-induced secretion of HSPA1L along with reversing cellular HSPA1L levels (Fig. 7c, d). We again confirmed that BSA-induced exocytosis of HSPA1L into culture medium was inhibited by rhVaspin (Fig. 7e), and concurrently HSPA1L was restored in cytoplasm by rhVaspin in HK2 cells (Fig. 7f).

**Vaspin is internalized via clathrin heavy chain (CHC)/GRP78 and CHC/HSPA1L mediated endocytosis.** After incubation of HK2 cell with rhVaspin, we detected rhVaspin in cell lysates by

Western blot analysis, and intracellular rhVaspin levels were increased by overexpression of GRP78 or HSPA1L (Supplementary Fig. S8a, b). In addition, co-administrations of anti-GRP78 antibody or anti-HSPA1L antibody with rhVaspin in HK2 cells, vaspin protein levels of cell lysate were decreased (Supplementary Fig. S8c). Next, we observed that GRP78 forms a complex with CHC (Supplementary Fig. S8e). This finding is supported by a previous report, showing an interaction of CHC with GRP78[31]. CHC is integral part of clathrin, which is a major protein component of coated pits and involved in intracellular trafficking of receptors and endocytosis. Like GRP78, a complex formation of HSPA1L with CHC was also observed (Supplementary Fig. S8f), and HSPA1L was detected in cell surface proteins as the case with GRP78 (Fig. 5c). However, the increase in cell surface HSPA1L by TM and TG was mild and the treatment with palmitate rather decreased cell surface HSPA1L (Fig. 5c). Next, vaspin protein levels in HK2 cells, overexpressed with increased doses of pMyc-CHC and incubated with rhVaspin, were increased in a CHC dose-dependent manner (Supplementary Fig. S8d). In vivo analysis, rhVaspin was injected intraperitoneally into vaspin−/−

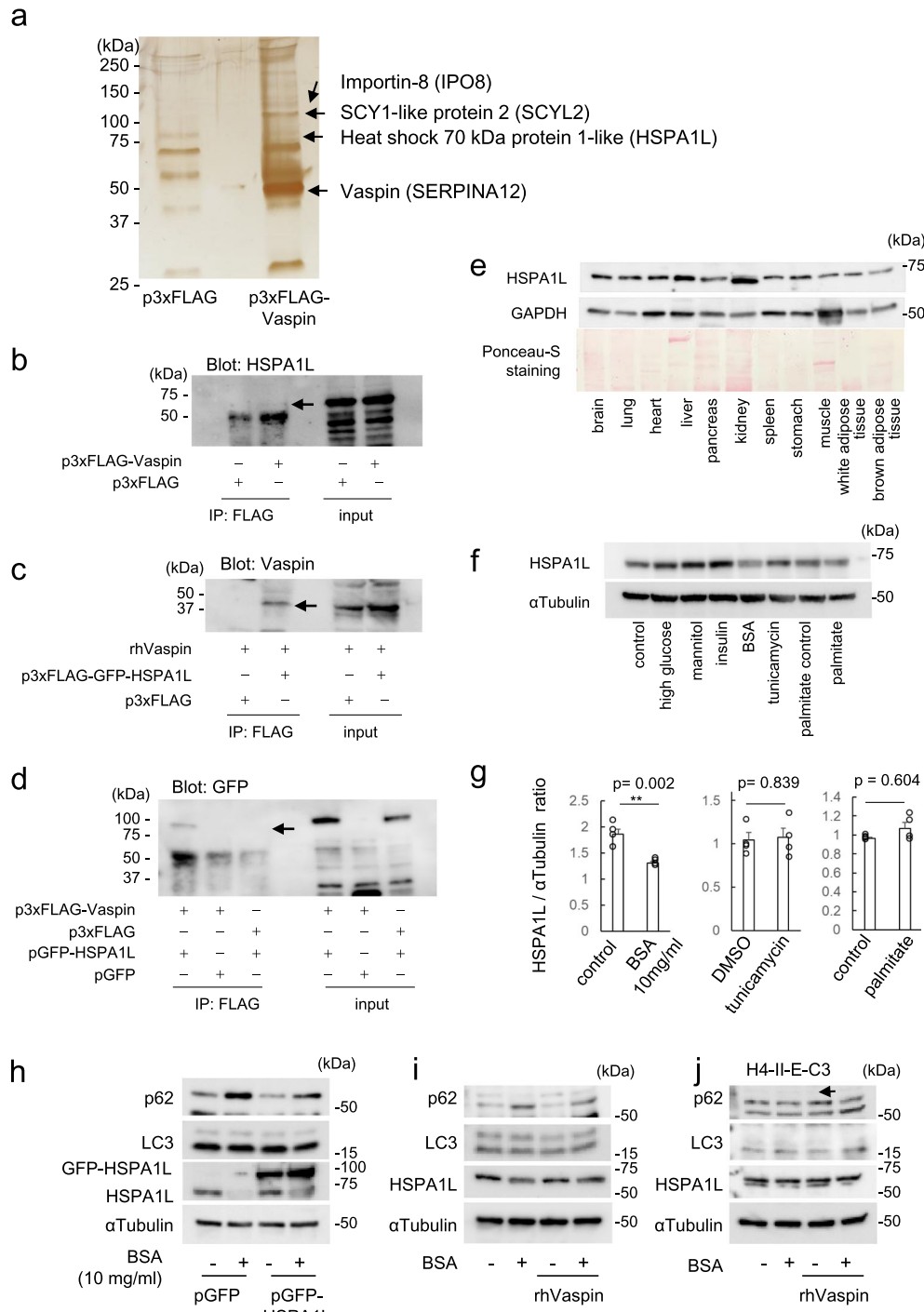

mice, and 30 min after injection, kidney samples were collected and subjected into further studies. In immunofluorescence staining, co-localization of rhVaspin with lamp2 was observed (Fig. 8). In fractionated kidney samples of vaspin−/− injected with rhVaspin, rhVaspin distributed from plasma membrane fractions to lysosomal fractions, ER fractions and mitochondrial fractions (Fig. 9).

Therefore, it was plausible that vaspin is internalized through CHC/GRP78 and CHC/HSPA1L mediated endocytosis in PTCs, and transported to organelle including ER and lysosome, where it acts on keeping ER stress adaptation, promoting autophagy and inhibiting LMP.

**Enlarged lysosomes of proximal tubular cells in humans with obesity and diabetes mellitus.** In human renal biopsy samples from diabetic nephropathy patients, enlarged lysosomes were visible by immunofluorescence staining of lamp1 (Supplementary Fig. S9a–c), while a patient with thin basement membrane disease (TBMD) as a nondiabetic and non-obese control demonstrated lamp1-positive lysosome as small dots (Supplementary Fig. S9d). In an overweight patient with TBMD of body mass index (BMI) 26.7 kg/m$^2$, lamp1 positive vacuole were observed in proximal tubules (Supplementary Fig. S9e). In diabetic nephropathy (DN) or obesity related kidney disease, lamp1 positive enlarged vacuoles were clearly visible, indicating enlarged lysosomes (Supplementary Fig. S9a–c, f).

**Fig. 6 HSPA1L as a vaspin-interactive molecule in HK2 cells. a** Immunoprecipitation of HK2 cells overexpressing p3xFLAG-Vaspin. Cell lysates were immunoprecipitated with ANTI-FLAG M2 affinity agarose gel and loaded into SDS-PAGE. LC-MS/MS analysis of excited bands revealed some candidate molecules interacting with vaspin. **b** Immunoprecipitation using HEK293T cells expressing p3xFLAG-Vaspin with ANTI-FLAG M2 affinity agarose gel. Blot with anti-HSPA1L antibody demonstrates complex formation of 3xFLAG-Vasin with HSPA1L. **c** HK2 cells expressing p3xFLAG-GFP-HSPA1L were cultured with recombinant human vaspin protein (rhVaspin). Cell lysates were immunoprecipitated with ANTI-FLAG M2 affinity agarose gel and blotted with anti-Vaspin antibody. rhVaspin was internalized and interacted with HSPA1L of HK2 cells. **d** Cell lysates of HEK293T cells expressing p3xFLAG-Vaspin and pGFP-HSPA1L were immunoprecipitated with ANTI-FLAG M2 affinity agarose gel. Blot with anti-GFP antibody confirmed an interaction of 3xFLAG-Vaspin with GFP-HSPA1L. **e**. HSPA1L protein expression levels of various organs of C57BL/6 J mice are demonstrated by Western blot analyses. **f** Alteration of HSPA1L protein levels of HK2 cells by Western blotting. HK2 cells were cultured with 5 mM glucose (control), 25 mM glucose (high glucose), mannitol (osmotic control; 5.5 mM glucose and 19.5 mM mannitol), 100 nM insulin, 10 mg/ml bovine serum albumin (BSA), 1 μg/ml tunicamycin, 0 or 500 μM palmitate. Intracellular HSPA1L was decreased by BSA administration. **g**. Relative expressions of HSPA1L in HK2 cells in western blot. Cells were cultured with or without 10 mg/dl BSA, 1 μg/ml tunicamycin or 500 μM palmitate for 24 h, Values are normalized by αTubulin. $N = 4$ independent experiments. **p < 0.01. **h** HK2 cells overexpressing pGFP or pGFP-HSPA1L were cultured with 10 mg/ml BSA or 100 ng/ml rhVaspin for 24 h. Western blot analyses demonstrated that HSPA1L ameliorated BSA-induced p62 accumulation. **i** HK2 cells were incubated with 10 mg/ml BSA and/or 100 ng/ml rhVaspin for 24 h. Vaspin inhibited BSA-induced p62 accumulation. **j** Hepatic cell line, H4-II-E-C3 cells, were incubated with 10 mg/ml BSA and/or 100 ng/ml rhVaspin for 24 h. There were no changes in p62 protein levels. Western blots were repeated twice.

**HSPA1L and GRP78 expression of diabetic nephropathy and obesity model**. In in vivo analysis using human renal biopsy samples, HSPA1L was stained mainly in tubules rather than glomerulus (Supplementary Fig. S10). In the patients with minor glomerular abnormality (MGA) or TMBD demonstrating no proteinuria, HSPA1L was visualized in both AQP1-positive PTCs and AQP1-negative distal tubular cells (Supplementary Fig. S10, panels TBMD and MGA-1). However, in DN with overt proteinuria, immunoreactivity of HSPA1L was decreased in PTCs and HSPA1L in distal tubules was accentuated compared with PTCs (Supplementary Fig. S10, panels IgA+DN and DN-1). Immunohistochemical localization of GRP78 was observed in AQP1-positive PTCs, and there were not differences among TBMD, DN, and ORG patients (Supplementary Fig. S11). In murine kidney, GRP78 expression was upregulated in PTCs of HFHS diet mice compared with STD mice. PTCs with prominent vacuoles in vaspin−/− mice were clearly accompanied with upregulation of GRP78 (Fig. 10).

## Discussion

Recently, the concept of tubulocentric view of DKD has emerged[1,32]. Proximal tubular cell injury is critical for both tubular and glomerular functions, because PTCs attach to Bowman's capsule and the loss of PTCs leads to nonfunctioning atubular glomeruli[1,33]. Therefore, unraveling mechanisms of PTCs' damage is important to developing strategy for prevention and reversal of deterioration in renal function. There has been no obviously effective approach to prevent tubular cells loss from metabolic stresses. This is the first report clearly demonstrating that vaspin, one of adipokines, has protective roles in PTCs under obesity and diabetes via binding to GRP78 and HSPA1L, both are member of Hsp70 family[18].

Yamamoto et al. reported that high fat diet induced vacuolation of PTCs in mice as phospholipid accumulated lysosomes[15]. They demonstrated that palmitic acid impairs lysosomal acidification and degradation activity; then undigested excessive lipids accumulate in lysosome. Vacuolation of PTCs in vaspin−/− mice under HFHS diet was also identified as enlarged lysosome, and PTCs revealed autophagy impairment demonstrated by p62 accumulation. Large lysosomes are fragile and easy to rupture due to alteration of membrane tension[22]. Instability of lysosomes promotes cell death program. Lysosomal membrane integrity is maintained by several factors, such as antioxidants, anti-apoptotic Bcl-2 proteins, Hsp70, lamp1/2 and lipid composition of lysosomal membrane[34]. In contrast, ROS, p53, Bax, caspases and calpains impair lysosomal membrane integrity, then subsequently

induce LMP[34]. Partial LMP induces cell apoptosis, and massive LMP triggers necrosis[34].

We demonstrated that palmitate and ER stress inducers, such as tunicamycin and thapsigargin, promoted relocalization of cathepsin B from lysosome into cytosol shown by immunofluorescence staining and Western blot analysis. Leakage of cathepsin B into cytosol, which was induced by palmitate, was inhibited by rhVaspin administration with statistically significance. ER stress enhanced by tunicamycin and thapsigargin also increased leakage of cathepsin B into cytosol, although there were no statistically differences. Recently, it is reported that galectin-3 puncta assay is more sensitive to detect LMP[24]. Galectin-3 is recruited to the sites of endolysosomal leakage, and thus detecting an appearance of galectin-3 puncta is useful technique to detect LMP. In present assay, tunicamycin and thapsigargin induced LMP, that was inhibited by rhVaspin administration. Although previous report demonstrated that LMP was not detected by galectin-3 assay in 200 nM thapsigargin treated MCF7 cells[24], in our study, HK2 cells treated with 1 μM thapsigargin demonstrated LMP, and this discrepancy may be due to a differences of cell lines.

Previous report demonstrated that palmitate-derived crystals activated NLRP3 inflammasome and subsequent IL-1β release via lysosomal dysfunction[35]. They showed that specific inhibitor of cathepsin B completely abolished palmitate-induced IL-1β release in lipopolysaccharide primed primary macrophages[35]. Another report described that cathepsin B, that is released into cytosol, induces activation of NLRP3 inflammasome[27]. Indeed, in our study, NLRP3 inflammasome pathway was activated by administration of palmitate, tunicamycin and thapsigargin. Up to the present, a great deal of research indicated a crosstalk between ER stress and NLRP3 inflammasome activation via UPR-dependent and -independent pathways[36,37].

Both obesity and diabetes enhance ER stress responses in various organs; adipose tissues, livers, muscle, brain and pancreatic β cells, and kidney[4,38]. Prolonged or excessive ER stress causes cells failure to compensatory responses, which accelerate maladaptive unfolded protein response and apoptosis via ATF4-CHOP pathway[39]. NLRP3 inflammasome, which is one of the major innate immune signaling molecules, is also activated during the development of obesity, diabetes mellitus and atherosclerosis[35], and induces inflammation and cell death[27].

Metabolic disorders are strongly associated with organelle stresses, via adaptive mechanisms at early stage, such as ER stress, Golgi stresses, mitochondrial stresses and lysosomal stresses. In chronic stage, such as metabolic syndrome and diabetes, a destiny of cell is dependent on a balance between adaptation and maladaptation to organelle stresses. Therefore, regulating organelle

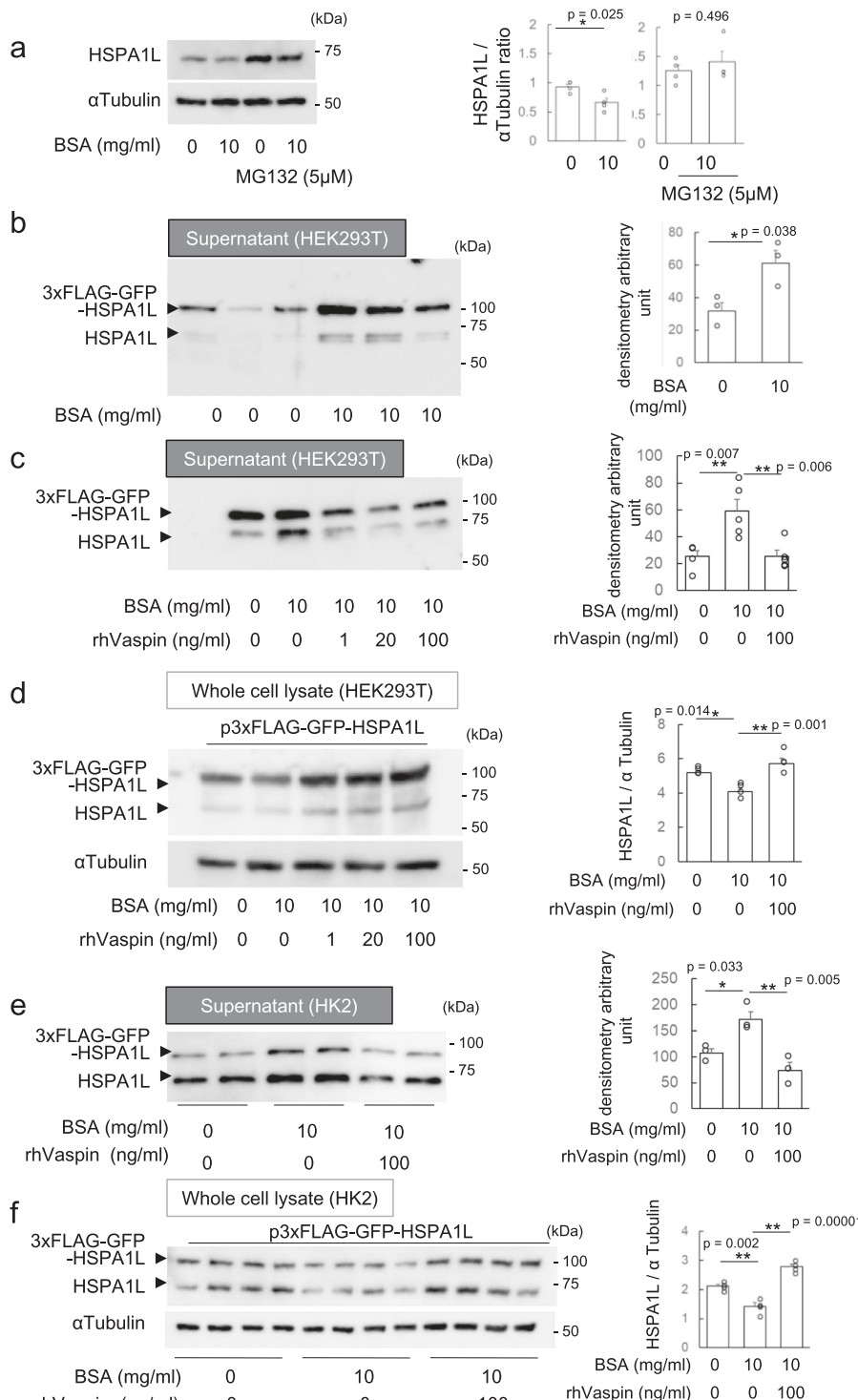

**Fig. 7 Inhibition of BSA-induced HSPA1L secretion by rhVaspin. a** HK2 cells were cultured with BSA 0 or 10 mg/ml at the presence of MG132 0 or 5 μM for 10 hr. BSA-induced decrease of HSPA1L protein levels was partially reversed by MG132. **b** HEK293T cells expressing p3xFLAG-GFP-HSPA1L were cultured with BSA at 0 or 10 mg/ml. Supernatant of HEK293T cell were immunoprecipitated with ANTI-FLAG M2 affinity agarose gel. Secreted HSPA1L was detected by Western blot analysis using anti-HSPA1L antibody. **c** HEK293T cells expressing p3xFLAG-GFP-HSPA1L were cultured with BSA at a dose of 0 or 10 mg/ml for 24 h. In addition to BSA, rhVaspin was administrated at a dose of 0, 1, 20, 100 ng/ml. Supernatant of HEK293T cell were immunoprecipitated with ANTI-FLAG M2 affinity agarose gel. Secreted HSPA1L was detected by Western blot analysis using anti-HSPA1L antibody. **d** Whole cell lysates were subjected to Western blot analyses. Blotting was performed by HSPA1L and normalized by αTubulin. **e** HK2 cells expressing p3xFLAG-GFP-HSPA1L were cultured with BSA at a dose of 0 or 10 mg/ml for 24 hr. rhVaspin was co-cultured at a dose of 100 ng/ml. Supernatants of HK2 cell were immunoprecipitated with ANTI-FLAG M2 affinity agarose gel. Secreted HSPA1L was detected by Western blot analysis using anti-HSPA1L antibody. **f** Whole cell lysates were subjected to Western blot analyses. Blotting was performed by HSPA1L and total densitometry quantification of HSPA1L and 3xFLAG-GFP-HSPA1L are normalized by αTubulin. Right panel shows statistical analysis. Data are shown as means ± S.E. $N = 4$ (**a**), 3 (**b**), 5 (**c**), 4 (**d**), 3 (**e**), 4 and (**f**), *$p < 0.05$, **$p < 0.01$. Western blots were repeated twice.

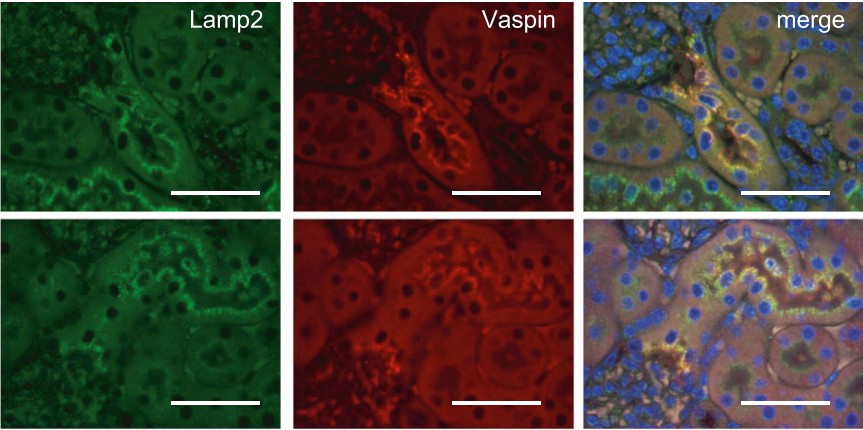

**Fig. 8 Internalization of rhVaspin into proximal tubular cells in vitro.** rVaspin was intraperitoneally injected into Vaspin−/− mice. After 30 min, kidney samples were collected, and subjected to immunofluorescence study. rhVaspin (red) was internalized into proximal tubular cells and co-localized with Lamp2 (green). Bar = 100 μm. Immunostaining experiments were repeated twice.

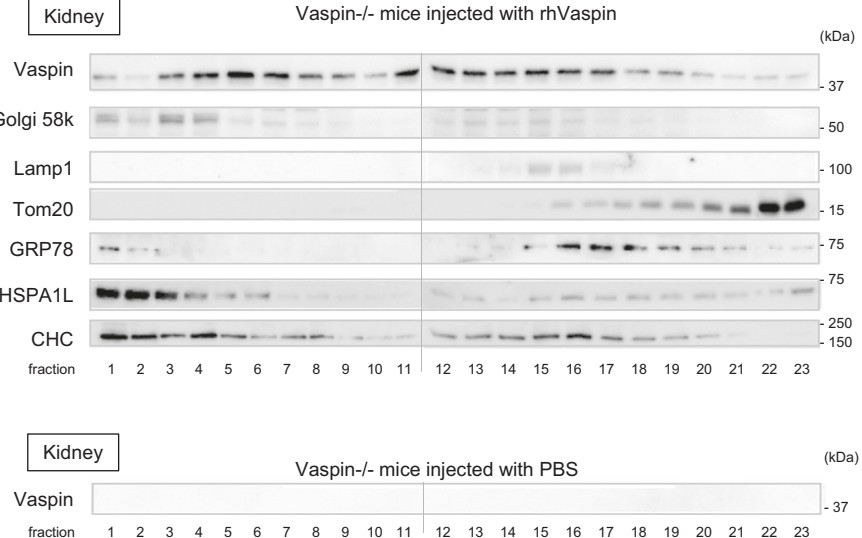

**Fig. 9 Cellular localization of rhVaspin injected into Vaspin−/− mice.** rhVaspin or PBS as a control were intraperitoneally injected into Vaspin−/− mice, and after 30 min, kidney samples were collected. Kidney tissues were ultra-centrifugated and fractionated by discontinuous iodixanol gradients method. We loaded the fractionated samples #1 to #23 on 2 separate SDS-PAGE gels and performed Western blot analysis. Two polyvinylidene fluoride membranes, fractions #1-#11 and #12-#23, were placed in this figure. rhVaspin protein was localized in throughout subcellular fractions. Western blots were repeated twice.

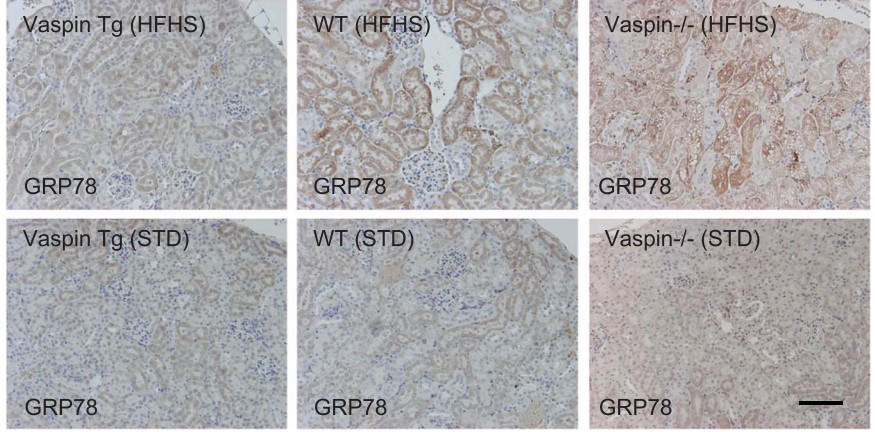

**Fig. 10 Immunohistochemical staining using kidney tissues of Vaspin Tg, WT and Vaspin−/− mice under high-fat high-sucrose (HFHS) or standard (STD) diet.** Bar = 100 μm. Immunoperoxidase staining studies were repeated twice.

stresses is important strategy to protect PTCs, and kidney function. We first demonstrated that vaspin protects PTCs from palmitate and ER stress induced LMP, NLRP3 inflammasone activation and subsequent cell death. Vaspin was identified from visceral adipose tissues of OLETF (Otsuka Long-Evans Tokushima Fatty) rat, an animal model of obese and type 2 diabetes mellitus[10], and we reported that vaspin inhibited obesity, insulin resistance, fatty liver[11] and vascular endothelial cell apoptosis[12]. In present study, we found that vaspin inhibited metabolic disorder-induced ER stress, autophagy impairment, disruption of lysosomal membrane integrity and subsequent NLRP3 inflammasome activation, and cell death in PTCs. Vaspin is a novel therapeutic target molecule in obesity related kidney diseases and DKD to the best of our knowledge. There are some limitations in our DKD model. Although the mesangial matrix expansion was observed in the HFHS diet-induced diabetes mice, the increase in albuminuria was not prominent compared to STD mice. The current model is characterized by prominent pathological changes in tubules and to a lesser extent in glomeruli. Thus, the current animal model may not be appropriate to evaluate an effect of vaspin on glomeruli. In a further experiment, the evaluation of DKD models with severe glomerular changes such as streptozotocin-induced diabetic model with prominent hyperglycemia to elucidate the roles of vaspin on diabetic glomerulosclerosis.

We also investigated molecular mechanisms of vaspin in protecting PTCs from metabolic stresses and identified an important role of vaspin/HSPA1L-mediated pathways in the kidney. Firstly, we demonstrated a role of HSPA1L in obesity and DKD. We identified HSPA1L as a vaspin-interactive molecule in PTCs and also found that HSPA1L forms a complex with lamp2 and promotes chaperon mediated autophagy (CMA) with subsequent inhibition of p62 accumulation under enhanced ER stress or palmitate induced lipotoxicity. This finding is supported by other study that Hsc70 (HSPA8), which is higher identical to HSPA1L[18,40], interacts with lamp2, the receptor for CMA[9,41]. We confirmed that vaspin cooperated with HSPA1L and inhibited p62 accumulation in PTCs. Interestingly, protein levels of HSPA1L in PTCs were significantly decreased by BSA administration. The depletion of HSPA1L in PTC was accompanied by extracellular secretion of HSPA1L. The current investigation unraveled the mechanism; the more albuminuria, the more reabsorbed albumin into PTCs, which deplete intracellular HSPA1L and induced subsequent autophagy impairment and disruption of homeostasis in PTCs. In human renal biopsy samples of the patients with overt proteinuria, HSPA1L was strongly expressed in distal tubules compared to proximal tubules. However, we failed to detect urinary HSPA1L excretion in mice by Western blot analysis, therefore one of the limitations of current study is that albumin-induced HSPA1L secretion is not evidenced by in vivo study. The development of high-sensitive ELISA system to detect HSPA1L would confirm this hypothesis. Jheng. HF et al. reported that albumin induces the release of HSP70 from proximal tubular cells, and HSP70 triggers the production of inflammatory mediators in a TLR4-dependent pathway[42]. Although they didn't described detail of HSP70 isoforms, major isoform of HSP70 is HSPA1A and HSPA1B, their alternative names; Hsp70, Hsp72 or Hsp70-1. In dendritic cells, HSPA1L is reported to activate Th1 response via direct binding to TLR4[43]. In this study, we focused on HSPA1L (Hsp70-1t), which shares 91% homology to HSPA1A, thus it is probable that secreted extracellular HSPA1L is pro-inflammatory on peripheral PTCs and downstream distal tubules in contrast intracellular HSPA1L, which is cytoprotective. Recently, urinary Hsp72: member of Hsp70s family is reported as a sensitive marker of acute kidney injury and prognostic indicator of chronic kidney disease of cats[44]. Further investigations are needed to explore the roles of Hsp70s family in the kidney. Specifically, we are planning to measure of human urinary HSPA1L in patients with acute or chronic kidney diseases including DKD and demonstrate a clinical significance of HSPA1L.

Secondly, GRP78 and HSPA1L form complexes with CHC respectively and involve in endocytosis. Especially, dynamic change of GRP78 expression on plasma membrane is quite interesting in terms of an internalization of GRP78 ligands or receptor-mediated intracellular signal transduction. While cell surface GRP78 was dramatically increased by tunicamycin, thapsigargin and palmitate, vaspin protein contained in cell lysates was not altered. This finding indicates that intracellular vaspin levels are regulated by several mechanisms. Further investigations about a fluidity of membrane trafficking and intracellular dynamics of vaspin are needed. There has not been known about the macromolecules, that are internalized and protect PTCs from metabolic stress-induced cell death up to now. The present study demonstrates that vaspin is a GRP78/HSPA1L ligand and a novel molecule protecting PTCs to the best of our knowledge. Cell surface GRP78 and HSPA1L of PTCs may have a role of receptor and transducing some signaling, although we couldn't find any other ligands than vaspin in PTCs. Further investigations about intracellular signaling via cell surface GRP78 and HSPA1L as receptors are needed. It is reported that apoptosis was detected in proximal and distal tubular cells, while not present in glomerular cells in human early DKD[45]. Therefore, prevention of proximal tubular cell death is the first step of inhibiting progression of DKD.

In conclusion, vaspin inhibited obesity- and diabetes-related ER stress enhancement, autophagy impairment, LMP, NLRP3 inflammasome activation in PTCs and subsequent cell death. Vaspin internalized via GRP78/CHC and HSPA1L/CHC associated endocytosis, then is transported to each organelle, where protect organelle from metabolic stress-induced maladaptation. Intracellular HSPA1L, which promote CMA of PTCs, was decreased along with increased extracellular secretion (Supplementary Fig. S12). The present study demonstrates unique mechanism of albuminuria induced PTCs injury mediated by HSPA1L, and we declare that vaspin/HSPA1L pathways potentially become therapeutic targets for protecting proximal tubular cells in diabetes and obesity related kidney diseases.

## Methods

**Animal experiment.** Vaspin transgenic C57BL/6J male mice under aP2 promoter and Vaspin knockout (Vaspin−/−) mice[11] were housed under 12-h light-dark cycle and had free access to water. Eight-week-old mice were subjected to standard diet (STD) group or high fat-high sucrose diet (HFHS) (5.56 kcal/g; fat 58%, carbohydrate 25.5%, and protein 16.4%) (D12331: Research Diet) group. At 30-week old, we obtained kidney and subjected to experiments. All animal experiments were approved by the Animal Care and Use Committee of the Department of Animal Resources, Advanced Science Research Center, Okayama University (OKU-2013425, OKU-2014088, OKU-2016051, OKU-2016356, OKU-2019055).

**Cell culture.** HK2 cells were cultured in DMEM with low glucose (5.5 mM glucose), GlutaMAX™ Supplement and pyruvate (GIBCO), and supplemented with 10% feral bovine serum (FBS), 100 U/ml penicillin and 100 μg/ml streptomycin at 37 °C in 5% $CO_2$. HEK293T cell were cultured in DMEM with high glucose (25 mM glucose), GlutaMAX™ Supplement (GIBCO), and supplemented with 10% FBS, 100 U/ml penicillin and 100 μg/ml streptomycin. H4-II-E-C3 cells (ECACC) were cultured in MEME (Minimum Essential Medium Eagle) containing 2 mM glutamine, 1% nonessential amino acids and 10% FBS. For the knockdown experiments, the HK2 cells were transfected with 5 MOI (multiplicity of infection) of MISSION shRNA lentivirus transduction particles for GRP78 (NM_005347) (shRNA-GRP78) or Non-Target shRNA control lentivirus transduction particles (shRNA-CON). Lipofectamine LTX Reagent (Thermo Fisher Scientific) was used for transient transfection in HK2 and H4-II-E-C3 cells, and Polyethylenimine Max (Polysciences, Inc.) in HEK293T cells. To examine an effect of overdose albumin on culture cells, Bovine Serum Albumin (BSA) was supplemented into culture medium, in which FBS was reduced to 1%.

**TdT-mediated dUTP nick end-labeling (TUNEL) assay**. MEBSTAIN Apoptosis TUNEL Kit III (Medical & Biological Laboratories, Nagoya, Japan) was used for TUNEL assay in culture cells. DeadEnd™ Colorimetric TUNEL System (Promega, WI, USA) was used for TUNEL assay in mouse kidney tissues.

**Materials**. Tunicamycin and thapsigargin were purchased from SIGMA, and the stock solution was prepared in dimethyl sulfoxide (DMSO) for the induction of ER stress. Palmitate (SIGMA) was prepared in DMSO then it was diluted to 5 mM with 5% BSA (fatty acid free) (SIGMA)-contained PBS and sonicated at 50 °C until complete dissolution for stock solution. As a negative control of palmitate, PBS containing 1% DMSO and 5% BSA was used. MG-132 (Enzo Life Sciences, Inc.) was diluted in DMSO and added to medium at final concentration of 5 μM.

**Recombinant protein**. We prepared recombinant human vaspin protein (rhVaspin) as described previously[12]. In brief, (His6)-tagged human vaspin was expressed in Escherichia coli using a pET expression system (pET16b, Novagen), and was purified. Then (His6)-tag was removed by Factor-Xa digestion and endotoxin was removed.

**Western blot analysis**. Kidney cortex tissues were excised and homogenized with lysis buffer (20 mM Tris-HCl, pH 7.4, 100 mM NaCl, 10 mM benzamidine-HC, 10 mM ε-amino-n-caproic acid, 2 mM phenylmethylsulfonyl fluoride and 1% Triton X-100). After centrifugation at 14,000 rpm for 30 min at 4 °C, the supernatants were collected for further analyses. For fractionation of kidney tissues, discontinuous iodixanol gradients method was applied the protocol S36 downloaded on May 5, 2008 (OptiPrep; Invitrogen). Cell surface proteins of HK2 cells were biotinylated and isolated by Pierce Cell Surface Protein Isolation Kit (Thermo Scientific). Lysosomal/membrane-cytosolic fractionation of HK2 cells were performed as described previously[46]. Equal amount of protein was subjected to SDS-PAGE under reducing conditions, and electroblotted onto Hybond P polyvinylidene fluoride membranes (GE Healthcare Life Sciences). The membranes were immersed in blocking solution containing 5% nonfat dry milk and Tris-buffered saline with Tween-20 (0.05% Tween-20, 20 mM Tris-HCl, and 150 mM NaCl, pH 7.6) or in 2% BSA and TBS with Tween-20. Then, the membranes were incubated with primary antibodies; rabbit monoclonal anti-BiP (C50B12), #3177, 1:1000; anti-phosphor-eIF2α (Ser51) (D9G8), #3398, 1:1000; anti-ATF4 (D4B8), #11815, 1:1000; anti-GAPDH (14C10), #2118, 1:1000; rabbit polyclonal anti-eIF2α, #9722, 1:1000; anti-Bax, #2772, 1:1000; mouse monoclonal anti-CHOP (L63F7), #2895, 1:1000; anti-αTubulin (DM1A), #3873, 1:1000; anti-Myc-Tag (9B11), #2276, 1:1000 (Cell Signaling Technology); mouse monoclonal anti-Cathepsin B (CA10), ab58802, 1:400; anti-LAMP2 (H4B4), ab25631, 1:500; rabbit polyclonal anti-NLRP3, ab214185, 1:1000; anti-caspase 1, ab17820, 1:1000; anti-clathrin heavy chain, ab21679, 1:1000; anti-SERPINA12, ab58975, 1:1000; anti-HSPA1L, ab154403, 1:1000; anti-GFP, ab290, 1:2000; anti-LAMP1, ab24170, 1:1000; anti-IL 1 beta, ab9722, 1:2500 (abcam); rabbit polyclonal anti-LC3, PM045, 1:1000; anti-p62 (SQSTM1), PM045, 1:1000 (MBL); rat monoclonal anti-GRP78 (76-E6), sc-13539, 1:200 (Santa Cruz Biotechnology); mouse monoclonal anti-FLAG M2, F1804, 1:1000 (SIGMA). They were then incubated with anti-rabbit or anti-mouse IgG conjugated with horseradish peroxidase (HRP), NA934 and NA931, 1:10000 (GE Healthcare Life Sciences) or anti-goat IgG conjugated HRP, sc-2020, 1:1000 (Santa Cruz Biotechnology), or anti-rat IgG conjugated HRP, #7077, 1:3000 (Cell Signaling Technology), or Rabbit TrueBlot: anti-Rabbit IgG HRP 1:1000 (ROCK-LAND). The blots were washed three times with Tris-buffered saline with Tween-20, immersed in ECL Plus Western Blotting Detection Reagents (GE Healthcare Life Sciences), and the chemiluminescence was analyzed using the LAS-4000 mini instrument (FUJIFILM).

**Immunoprecipitation**. Protein lysates were precleared using sepharose 4B (SIGMA) at 4 °C for 1 h to remove non-specific binding protein. After that, immunoprecipitation was performed using Catch and Release v2.0 (Millipore) for HK2 cells and kidney protein. Immunoprecipitation using FLAG tag fused protein were performed by ANTI-FLAG M2 Affinity Gel (SIGMA).

**Plasmid**. Plasmids of pEGFP-hGal3 (pGFP-galectin-3) and pcDNA5/FRT/TO GFP HSPA1L (pGFP-HSPA1L) were gifts from Tamotsu Yoshimori (Addgene plasmid #73080) and Harm Kampinga (Addgene plasmid #19484), respectively. cDNA of 3xFLAG was amplified by using PCR primers (3xFLAG-F and 3xFLAG-R) and ligated to BamHI site of pcDNA/FRT/TO GFP HSPA1L (p3xFLAG-GFP-HSPA1L). SERPINA12 (vaspin) cDNA without signal peptide was amplified by using PCR primers (Flag-EcoRI-hVaspin-F and OL64-pTNT-AS) and ligated to EcoRI-XbaI site of p3xFLAG CMV expression vector (SIGMA) (3xFLAG-Vaspin). Coding sequence without signal peptide of GRP78 (19–654) was amplified by using PCR primers (Flag-HindIII-GRP78-S and GRP78-AS1-RE) and subcloned into TOPO TA Cloning vector (Invitrogen). Then, the plasmid was digested with HindIII/XbaI and ligated to pEGFP-C3 vector (Clontech) (p3xFLAG-GRP78). Primers are shown in Supplementary Table S1. pcDNA3 with myc (myc) and pcDNA3 carrying clathrin heavy chain with myc-tag (pMyc-CHC) are prepared as reported previously[47].

**Immunofluorescence microscopy**. Cells were cultured on coverslips and fixed in 100% methanol for 15 min, permeabilized with 0.1% Triton X-100 in PBS for 15 min, blocked with serum-free protein block (DAKO) for 30 min. Human kidney samples were also blocked with serum-free protein block (DAKO) for 30 min. Then these were incubated overnight with primary antibodies; rat monoclonal anti-LAMP2 [GL2A7], ab13542, 1:500; rabbit polyclonal anti-HSPA1L, ab154403, 1:400 (abcam); rabbit polyclonal anti-LAMP1, ab24170, 1:300; rabbit monoclonal anti-Cathepsin B (D1C7T), #31718, 1:800 (Cell Signaling Technologies); rabbit monoclonal anti-p62 (SQSTM1) antibody, PM045, 1:600 (BML); mouse monoclonal anti-AQP1 (B11), sc-25287, 1:300 (Santa Cruz Biotechnology). After washing, cells were incubated with second antibodies; Alexa Fluor 488, 555, or 564 for 60 min at room temperature (A-21200, A-21208, A-21441, A11005 1:500, Invitrogen and #4413 1:500, Cell Signaling Technologies). After that mounted by ProLong™ Diamond Antifade Mountant with DAPI (Thermo Fisher Scientific) and were imaged using a microscope (OLYMPUS). Lysosome of HK2 cells was stained with LysoTracker Red DND-99 (Invitrogen).

Quantification of lysotracker staining of HK2 cells was performed by Image J (https://imagej.nih.gov/ij/docs/guide/146.html). In brief, the images were split into red, green, and blue channels and a threshold was set in red channel between 155 and 255 to isolate and separate the individual lysosomes. We evaluated the average size of particles with more than 10 pixel$^2$ to evaluate the enlarged lysosomes. For quantification of Lamp1, we set threshold in green channel between 190 and 255, and evaluated average size of particles with more than 10 pixel$^2$. Area of p62 staining was also analyzed in red channel, in which a threshold was set between 120 and 255.

**Morphological studies**. Kidney tissue specimens were fixed in 10% formaldehyde, embedded in paraffin, and 4 μm-thick sections were prepared. They were deparaffinized, rehydrated and pretreated by microwave for 10 min in Target Retrieval Solution (DAKO). Endogenous peroxidase activity was blocked in 3% hydrogen peroxide. Nonspecific binding was blocked by incubation in 10% goat or rabbit serum for 30 min. The tissue sections were incubated with rabbit polyclonal anti-LAMP1 ab242170 1:200, anti-HSPA1L ab154409 1:500 (abcam), anti-BiP (C50B12) #3177 1:200 (Cell Signaling Technology), anti-AQP1 (H-55) sc-20810 1:100 (Santa Cruz Biotechnology), rabbit monoclonal anti-p62 (SQSTM1) PM045 1:1000 (MBL), at 4 °C overnight. After being washed in PBS, they were incubated with a biotinylated secondary antibody and VECTASTAIN ABC Standard Kit (Vector Laboratories, Burlingame, CA). Immunochemical staining was performed with the ImmPACT DAB SUBSTRATE (Vector Laboratories).

Vacuolation and tubular dilatation were scored between 0 and 5 according to the following scoring system: 0 = 1<% cortex affected; 1 = 1–10% cortex affected; 2 = 11–25% cortex affected; 3 = 26–50% cortex affected; 4 = 51–74% cortex affected; 5 ≥ 75% cortex affected. The mesangial matrix index (MMI), which was measured using Image J, was defined as the PAS-positive area in the tuft area, calculated using the following formula: MMI = (PAS positive area)/(tuft area).

**LC-MS/MS (high performance liquid chromatography-tandem mass spectrometry)**. We performed immunoprecipitation with HK2 cells lysates over expressed p3xFLAG-Vaspin or p3xFLAG using ANTI-FLAG M2 Affinity Gel (SIGMA). The band, visible only in cell lysate overexpressed p3xFLAG-Vaspin, were excised and in-gel-digested with trypsin, subjected to LC-MS/MS, and analyzed with Mascot Ver. 2.5.1 and Scaffold Ver. 4.8.3.

**Detection of HSPA1L in supernatant of cultured cell**. HEK293T cells overexpressing p3xFLAG-GFP-HSPA1L was incubated with BSA for 24 h, then culture medium was collected. Collected medium was concentrated by centrifugal filter, Amicon® Ultra (Merck). Concentrated sample was precleared using sepharose at 4 °C for 1 h to remove non-specific binding protein, then immunoprecipitation using FLAG tag fused protein were performed by ANTI-FLAG M2 Affinity Gel (SIGMA). Eluted samples were loaded on SDS-PAGE and HSPA1L was detected by Western blot analysis.

**Human sample**. Frozen kidney sections obtained from clinical renal biopsies at Okayama University Hospital were used for immunofluorescence analysis. Informed consent was obtained. It was approved by Okayama University Graduate School of Medicine, Dentistry and Pharmaceutical Sciences and Okayama University Hospital, Ethics Committee (1709-039). We retrospectively selected samples with obesity related kidney disease and type 2 diabetic nephropathy, at the point of clinical data; HbA1c, proteinuria, BMI, and histological diagnosis.

**Statistics and reproducibility**. For parametric analyses, the multiple comparisons were performed by one-way ANOVA with Tukey–Kramer method and two group comparisons by unpaired two-tailed $t$ test using SPSS software (IBM, Chicago, IL). For non-parametric analyses, the multiple comparisons were performed by Kruskal–Wallis test with Bonferroni correction. All experiments were performed in duplicate to confirm reproducibility.

**Reporting summary**. Further information on research design is available in the Nature Research Reporting Summary linked to this article.

## Data availability

LC-MS/MS source data can be found in Supplementary Data 1. Source data behind all graphs can be found in Supplementary Data 2. The authors declare that all other data supporting the findings of this study are available upon reasonable request to the authors.

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

## Acknowledgements

We thank M. Matsuyama and M. Fukushima for technical and clinical suggestions. We acknowledge support from Central Research Laboratory, Okayama University Medical School; usage of BECKMAN COULTER XL 80, ABI PRISM3130, and producing paraffin blocks and sections. This work was supported by a Grant-in-Aid for Scientific Research (C) 26461361, Japan Diabetes Foundation, Grants for young researchers from Japan Association for Diabetes Education and Care, Mochida Memorial Foundation for Medical and Pharmaceutical Research, The Ichiro Kanehara Foundation for the Promotion of Medical Sciences and Medical Care, The Okayama Medical Foundation, Japan medical Women's Association, Suzuken Memorial Foundation, Yukiko Ishibashi Foundation, and SHISEIKAI Scientific Award to A.N.

## Author contributions

A.N., J.W. designed the project and experiments and wrote the manuscript. A.N., S.Y., and J.E. performed experiments and analyzed and interpreted data. S.K. and Y.I. generated knockout mice. H.S. designed clinical study using human renal biopsy samples.

## Competing interests

J.W. receives speaker honoraria from Astra Zeneca, Daiichi Sankyo, MSD, Novartis, Tanabe Mitsubishi, Taisho Toyama and receives grant support from Baxter, Chugai, Dainippon Sumitomo, Ono, Teijin. All other authors declare no competing interests.
