## [Peer Review File · Communications Biology]

Reviewers' comments:

Reviewer #1 (Remarks to the Author):

In this manuscript, Nakatsuka et al. describe the potential protective role of vaspin against tubular damage caused by HFHS and in the putative lipotoxic effect of palmitic acid. The authors have clear in vivo data that the lack of vaspin exacerbates the obesity-related tubular damage. The authors then show in vitro (in HK2 cells and HEK-293 cells) that damage produced by palmitic acid, the putative lipotoxic factor of the obesity-related metabolic disturbance, can be prevented by rhVaspin. PA in vitro provoked lysosome enlargement, cathepsin B release and apoptosis. The protective mechanism of Vaspin depends on its binding to GRP78 and HSPA1L. When the binding of these three proteins occurs, they are internalized in CHC clathrin-containing vesicles where they help mitigate ER stress, increase autophagy, etc.

While the concept is interesting, many aspects of this study require clarification and additional controls to make the claims convincing.

While the authors present novel data to show that 1. rhVaspin prevents PA-induced ER stress in vitro and 2. HSPA1L changes with albumin, there are several important concerns.

These can be summarized in three main areas:

1. Lack of fully characterizing the renal impairment and glomerular injury in their DKD model.
2. Lack of clear quantification of imaging studies and western blots, which causes concerns about technical aspects of the work.
3. Lack of in vivo validation of the mechanism by which vaspin protects from vacuolations and tubular damage (for example, HSPA1L is not shown in the mouse model).

Major concerns:

Figure 1: The authors aim to show that vaspin^{-/-} mice have an exacerbated damage in the proximal tubules. However, the figure does not fully convince the reader that this damage occurs only in proximal tubules, it appears much more diffuse. To address this, the authors should show the vacuolations in the AQP1 positive tubules and show this staining in all experimental groups, not only in WT (HFHS) and Vaspin^{-/-} (HFHS) (as it is currently in the supplementary figure). Also, it is concerning that in the supplementary IH of Vaspin^{-/-} (HFHS) mice, there are no vacuolations, how can the authors explain this inconsistency? Also, the toluidine-blue staining is not clear and also needs to be repeated across all experimental groups.

Importantly, the authors conclude from the first figure that "lysosomal enlargement in proximal tubular cells was enhanced in diet-induced obese vaspin^{-/-} mice, while vacuolar formation was "ameliorated in vaspin Tg mice." However, there is no clear evidence supporting this statement.

For a convincing characterization of tubular damage, the authors should include a tubular injury score which should contain a vacuolation score and a tubular dilation score. The latter is to make the findings quantifiable rather than only descriptive.

The authors report in their previous paper (Nakatsuka et al 2012) that their diet is enough to induce metabolic syndrome in mice. However, there is no characterization of the renal damage phenotype. The authors need to include measures of renal function, for example: urinary albumin to creatinine ratio, serum creatinine, serum BUN. Usually, these parameters have been shown to be altered in DKD models around week 30.

In addition to renal function parameters, it is well known that in any DKD model, there is some degree of glomerular damage. While their argument about the lack of understanding regarding tubular damage in DKD is true, it is also true that glomerular damage would be expected nevertheless, or there is some doubt that this mouse model truly represents a diabetic kidney model. The authors do not show any of the DKD hallmark histological findings like GBM thickening or mesangial expansion in any of their light microscopy. This leads the reader to question their model. Could the authors comment on why they don't show these findings or why they think their

model does not show glomerular damage?

The inconsistency in the cell types chosen for in vitro experiments is not justified. Could the authors explain why some experiments are done in HK2 cells and other experiments are done in HEK-293 cells, or better yet, repeat all experiments in one cell model?

Quantifications should be performed for all cell imaging (lysotracker, Lamp1 staining, anti-p62, anti-cathepsin B). Additionally, it is important that the authors show the images of the TUNEL quantification in Figure 2g.

In figure 4 the authors DO NOT demonstrate that IL-1b increase or apoptosis is specific of proximal tubules, there is no staining clarifying this, so this claim is not supported by the data.

Figure 4: Could the authors comment on why there is no statistical significance in Figure 4 between the WT (HFHS) and Vaspin Tg (HFHS)? Does this mean that Vaspin does not protect against apoptosis? Why?

Figure 6 and following: the authors say that they performed Anti-FLAG IP purification of transfected vaspin and then purified this band to depict interactors with LC-MS. From here they found the interaction with HSPA1L. The authors do not show the Proteomics data and data analysis, which should be shown at least in a supplementary figure, table etc.

The critical interaction between vaspin and HSPA1L should be shown in a native system: It is one thing to show the interaction of a protein in an overexpression system and another to show that this interaction is physiologically relevant. If the proposed mechanism through which vaspin protects and ameliorates tubular damage is by binding to HSPA1L and causing further internalization (e.g. preventing secretion of HSPA1L), then the authors must show this interaction in the rhVaspin treated mice, or the Vaspin Tgs.

Figure 6f and g: the quantification is based on GAPDH expression but the representative blot has anti-tubulin as a loading control. This is an example of the general disorganization and inconsistencies found throughout the manuscript and figures.

Also, in this experiment can the authors clarify if they add the BSA in the medium that is already supplemented with FBS or not? If it is, adding 10 mg/ml of BSA is equivalent to a 4+ proteinuria, which is massive proteinuria. If in addition to that the culture media has FBS too, I would be worried about the possibility of an osmotic stress that is not equivalent to the mannitol control.

Another inconsistent result: why does vaspin not prevent the secretion (in supernatant) of the overexpressed 3xFLAG-GFP-HSPA1L protein?

The finding that HSPA1L is found mainly in distal tubules of patients with proteinuria is not clear. The stainings should be improved. In addition, the fact that HSPA1L is present in the distal tubules rises a possibility of it being secreted by the distal tubule and not secreted to the distal tubule, it is not clear what is cause and effect. It is known that this protein is secreted by the collecting duct in response to environmental perturbations (see Pockley AG (2003) Heat shock proteins as regulators of the immune response. *Lancet*. 362:469–476). If the authors want to show definitive secretion, the presence of HSPA1L in the urine of mice should be evaluated, as well as the increase in urine in response to increased proteinuria. This finding should then be abrogated either in rhVaspin treated mice or in Vaspin Tg mice.

Minor concerns:

1. The authors mention in the main text and in the figure legends that they studied these mice at 30 weeks of age. However, the methods say they were sacrificed at 24-weeks. Did the authors study mice after this stage and fully characterize a CKD phenotype?
2. Please specify the time the cell cultures were incubated with rhVaspin.
3. Please provide quantification data for all Western blots. Conclusions should not be drawn from 1 experiment.
4. Please add the appropriate kDa reference to all western blots and not only in some of them.

5. In Figure 3b, there is clearly more protein loaded on the lane of the PA-treated cells, as evidenced by increased GAPDH. Therefore the representative Western blot is not truly representative. Please repeat the experiment and show the resulting representative blot.
6. In Figure 3e, the representative Western blot also needs improvement.
7. Figure 4e, please show TUNEL staining of control tissues too.
8. Could the authors specify the TUNEL kit assay that was used?
9. Quantification for Supplementary figure 5 would be desirable.
10. The arrangement of Figure 6d is difficult to read. Please rearrange this figure, for example, using + or - signs.
11. Figure 6e needs to be repeated and add the same amount of protein in each lane, otherwise the expression levels are not comparable.
12. Data analysis and Statistics: For studies shown in Figure 2, it looks like the comparisons have been designed to set the controls to one, and eliminate the variance of the control. This design makes statistical inferences prone to error. The data should be reanalyzed as in the other studies so that the variance of test and control groups is accounted for.

Reviewer #2 (Remarks to the Author):

This study had identified a novel role for the visceral adipose tissue derived serine protease inhibitor (Vaspin), an important adipokine, in the pathogenesis of diabetic kidney disease (DKD) through ameliorating ER stress, autophagy impairment and lysosome dysfunction in proximal tubular cells (PTCs). Using Vaspin^{-/-} KO mice in a high fat high sucrose diet (HFHS) model, they found prominently enlarged lysosomes in PTCs which was associated with increased cell apoptosis compared to that seen in their wild type litter mates. Furthermore, the authors also conducted mechanistic experiments in HK2 and HEK293T cell lines to test the role of vaspin in protecting PTCs from metabolic stress. Their findings indicate that intracellular vaspin levels are regulated by several molecular mechanism. Firstly, they found that heat shock protein family A (Hsp70) member 1 like (HSPA1L) as a new vaspin interactive protein in PTCs. Secondly, they found that GRP78, a cell-surface receptor, forms a complex with HSPA1L and clathrin heavy chain (CHC) respectively and are involved in endocytosis of vaspin. They conclude that vaspin may play a role as an effector of GRP78/HSPA1L signaling.

Overall, this manuscript includes new and interesting data showing unique cross talk between adipose derived adipokine and PTC function related to DKD. Vaspin appears to be a novel protective factor for proximal tubular cells in diabetes and obesity related DKD. The conclusions are derived from in vivo models, in vitro mechanistic data as well as some human relevant information. There are however several aspects to be addressed to strengthen the data and the manuscript.

Major points

1. The authors should provide some results about the phenotype of the mice. Please show if there are any difference in body weight, blood glucose level, ITT, GTT, and adipocytes hypertrophy between Vaspin Transgenic, Vaspin^{-/-} KO and wild type mice.
2. Kidney histology shows severe tubular damage in Vaspin^{-/-} KO mice therefore it would be more interesting to show kidney dysfunction i.e urine Albumin/creatinine in these mice
3. As Vaspin is an important adipokine to reduce ER stress induced by obesity, please explain and provide some results whether Vaspin^{-/-} KO STD mice developed any histopathological changes in adipose tissue and kidney compare to wild type mice.
4. Western blot image in figure 2 does not show significant increase of GRP78 in PA treated cells - a quantitative validation is needed.
5. Further experiments, i.e gene expression and IHC staining, should be performed to support the presence of HSPA1L in the mice kidney. Figure 6e is not solid evidence as the band is almost non-detectable.
6. What is the effect of ER stress on HSPA1L?

7. Vaspin Transgenic and Vaspin^{-/-} KO mice have been used in previous published paper by authors. However, for the current study a clear description of mice model should be provided.

8. Please mention HFHS diet components. How much calories come from fat /carbohydrate /sugar?

9. In the methods, authors have mentioned that 24-week-old kidneys were collected, but results indicate kidneys were from 30 week old mice? Please specify the exact number of weeks that the mice were under HFHS diet and when they were euthanized to collect kidney samples.

10. In the cell culture studies please mention the concentration of low and high glucose treatments.

11. Please provide the catalog number and concentrations of all primary and secondary antibodies used for IF,IHC, and western blot experiments

Here are some recommendations for the Figures:

Figure 1: more details and explanation of the results seen needs to be added to the figure legend.

- Show vacuoles and Lamp1 positive lysosomes using arrows.
- Mention that different magnification has been used for images g-i and j-l)

Figure 2:

- Please add markers for western blot images
- Please provide more specific details about statistical analysis if one way ANOVA followed by post hoc tests to explore differences between multiple groups.
- P values are missing from the legends

Figure 3:

- Please follow chronological order when writing legends for this figure i.e. a-d.

Figure 4:

- Western blots – please add Molecular weight markers to indicate which band has been quantified.
- Please add the names of NLRP3 inflammasome pathway proteins in the legend.

Figure 6:

- Western blot pictures need marker to detect bands.

Figure 1 supplementary:

- Please follow chronological order when writing legends for this figure i.e. a-d.
- Indicate lysosomes using arrow in electron micrographs.

Fig 2 supplementary:

- Western blot pictures need molecular weight markers.
- Please say the white arrows indicate positive apoptotic cells in the legends.
- P values are missing in the figure legends.

Figure 8 supplementary:

- The legend is not very clear and the figure needs more clear labeling.

Figure 9 supplementary:

- Please provide key to the abbreviation on the figures to improve clarity
- Include the complete name for U-pro in the legend.
- Rearrange the images based on the order of results.
- Mention figure number in the Results text accordingly.

Responses to reviewer #1

While the authors present novel data to show that 1. rhVaspin prevents PA-induced ER stress in vitro and 2. HSPA1L changes with albumin, there are several important concerns.

These can be summarized in three main areas:

1. Lack of fully characterizing the renal impairment and glomerular injury in their DKD model.
2. Lack of clear quantification of imaging studies and western blots, which causes concerns about technical aspects of the work.
3. Lack of in vivo validation of the mechanism by which vaspin protects from vacuolations and tubular damage (for example, HSPA1L is not shown in the mouse model).

Major concerns:

1. Figure 1: The authors aim to show that vaspin^{-/-} mice have an exacerbated damage in the proximal tubules. However, the figure does not fully convince the reader that this damage occurs only in proximal tubules, it appears much more diffuse. To address this, the authors should show the vacuolations in the AQP1 positive tubules and show this staining in all experimental groups, not only in WT (HFHS) and Vaspin^{-/-} (HFHS) (as it is currently in the supplementary figure). Also, it is concerning that in the supplementary IH of Vaspin^{-/-} (HFHS) mice, there are no vacuolations, how can the authors explain this inconsistency? Also, the toluidine-blue staining is not clear and also needs to be repeated across all experimental groups.

Importantly, the authors conclude from the first figure that “lysosomal enlargement in proximal tubular cells was enhanced in diet-induced obese vaspin^{-/-} mice, while vacuolar formation was “ameliorated in vaspin Tg mice.” However, there is no clear evidence supporting this statement.

For a convincing characterization of tubular damage, the authors should include a tubular injury score which should contain a vacuolation score and a tubular dilation score. The latter is to make the findings quantifiable rather than only descriptive.

We thank reviewer #1 and completely agree with the critical comments. We add images of AQP1 staining of all groups in **Supplementary Fig. 1a**. We show the representative images of Vaspin^{-/-} (HFHS) with prominent vacuolations. The toluidine-blue staining is also newly performed in all groups. In **Supplementary Fig. 1b**, the images reveal the lysosomal enlargement in proximal tubular cells was accentuated in diet-induced obese Vaspin^{-/-} mice,

while vacuolar formation was ameliorated in Vaspin Tg mice. Finally, vacuolation and tubular dilatation were scored and quantified in **Supplementary Fig. 1d and 1e**.

2. *The authors report in their previous paper (Nakatsuka et al 2012) that their diet is enough to induce metabolic syndrome in mice. However, there is no characterization of the renal damage phenotype. The authors need to include measures of renal function, for example: urinary albumin to creatinine ratio, serum creatinine, serum BUN. Usually, these parameters have been shown to be altered in DKD models around week 30.*

In addition to renal function parameters, it is well known that in any DKD model, there is some degree of glomerular damage. While their argument about the lack of understanding regarding tubular damage in DKD is true, it is also true that glomerular damage would be expected nevertheless, or there is some doubt that this mouse model truly represents a diabetic kidney model. The authors do not show any of the DKD hallmark histological findings like GBM thickening or mesangial expansion in any of their light microscopy. This leads the reader to question their model. Could the authors comment on why they don't show these findings or why they think their model does not show glomerular damage?

We thank reviewer #1 for the important comment. As Reviewer #1 pointed out, we agree that the measures of renal function may be altered in DKD mouse models around week 30. The WT mice under HFHS chow demonstrated increased albumin excretion compared with WT mice under STD chow, however, it did not reach statistically significant levels (**Supplementary Fig. 1h**). We did not observe apparent changes of urinary albumin excretion in Vaspin Tg and Vaspin^{-/-} mice fed with HFHS chow compared with WT. In addition, we preliminary measured serum creatinine levels in part of the animals, but there were no changes. We add the urinary albumin excretion data in **Supplementary Fig. 1h** and describe in Result section, page 6 lines 1-12, "*In Vaspin Tg, wild type (WT) and Vaspin^{-/-} mice under STD, the vacuolations in tubules were absent, while HFHS diet significantly increased the vacuolation score. Comparison among Vaspin Tg, WT, Vaspin^{-/-} mice under HFHS diet demonstrated that the score of Vaspin Tg mice was significantly lower than WT and Vapsin^{-/-} mice (Supplementary Fig. 1d). There were no statistical differences of dilatation score among Vaspin Tg, WT and Vaspin^{-/-} mice under STD or HFHS diet (Supplementary Fig. 1e). As regards the glomerular legion, HFHS diet-induced increase of mesangial matrix index was inhibited in Vaspin Tg mice compared with WT and Vapsin^{-/-} mice (Supplementary Fig. 1f and 1g). Urinary albumin tended to increase in the mice fed with HFHS diet compared with STD, although there were no statistical differences. Under HFHS diet, there were no differences of urinary albumin among Vaspin Tg, WT and Vapsin^{-/-} mice (Supplementary Fig. 1h).*"

One of the possible causes for the lack of the significant alterations in albumin excretion is that the current diet-induced obesity model using C57BL/6J is resistant to glomerular injury (Wickes SE. *Biochimie*. 124: 65-73, 2016). We are now working on evaluate the glomerular injury and renal dysfunction in streptozotocin (STZ)-induced diabetic model for next project. Prominent hyperglycemia and oxidative stress are known to contribute to a pathogenesis of STZ-induced diabetic glomerulopathy (**Figures for reviewers, Fig. 1-3**). The current diet-induced obesity model using C57BL/6J is characterized by the lipid toxicity and ER stress-induced tubular injury model and there is a limitation for the investigation of diabetic glomerulopathy. Thus, we discuss about the limitation in page 15 lines 1-8, *“There are some limitations in our DKD model. Although the mesangial matrix expansion was observed in the HFHS diet-induced diabetes mice, the increase in albuminuria was not prominent or significant compared to STD mice. The current model is characterized by prominent pathological changes in tubules and to a lesser extent in glomeruli. Thus, the current animal model may not be appropriate to evaluate an effect of vaspin on glomeruli. In a further experiment, the evaluation of DKD models with severe glomerular changes such as streptozotocin-induced diabetic model with prominent hyperglycemia to elucidate the roles of vaspin on diabetic glomerulosclerosis.”*

3. *The inconsistency in the cell types chosen for in vitro experiments is not justified. Could the authors explain why some experiments are done in HK2 cells and other experiments are done in HEK-293 cells, or better yet, repeat all experiments in one cell model?*

As reviewer #1 pointed out, we used both HK2 and HEK293T cells. Most of the *in vitro* experiments were done in HK2 cells since they are the proximal tubular cell lines. For pull-down assay and immunoprecipitation (IP) to check the protein-protein interactions, more amount of protein with scale-up of cell culture is required. For the pull-down assay and IP appeared in **Fig. 5b, 6c, Supplementary Fig. 7, 8e and 8f**, HEK293T cells are used since HEK293T cells transfected with Polyethylenimine Max (PEI MAX) required much lower costs compared with Lipofectamine LTX. Unfortunately, HK2 cells are successfully transfected with Lipofectamine LTX, but not with Polyethylenimine Max (PEI MAX). We specifically describe the transfection methods in page 20 lines 1-2, *“Lipofectamine LTX Reagent (Thermo Fisher Scientific) was used for transient transfection in HK2 and H4-II-E-C3 cells, and Polyethylenimine Max (Polysciences, Inc.) in HEK293T cells.”*

However, in the experiments of protein secretion into supernatants, we completely agree with reviewer #1. In **Fig. 7**, we repeated the experiments in HK2 cells, which were done in HEK293T.

4. Quantifications should be performed for all cell imaging (lysotracker, Lamp1 staining, anti-p62, anti-cathepsin B). Additionally, it is important that the authors show the images of the TUNEL quantification in Figure 2g.

We appreciate reviewer #1 for pointing out our insufficient analyses. We performed quantification for cell imaging including lysotracker (**Fig. 2a**), anti-Lamp1 (**Fig. 2b**), and anti-p62 staining (**Fig. 2f**). Pixel area of each particle was measured using Image J and method is described in page 23 lines 14-21. The images of TUNEL assay is added in **Fig 2e**. We describe in page 6 lines 17-24, "We focused on palmitate-induced lipotoxicity in lysosomes of PTCs. In HK2 cells, lysosome was stained (**Fig. 2a**) and lysosomal membrane marker lamp1 (**Fig. 2b**). Palmitate administration promoted lysosome enlargement. The expressions of phosphorylated eIF2 α , GRP78 and C/EBP homologous protein (CHOP) were increased by palmitate administration and inhibited by recombinant human vaspin protein (rhVaspin) (**Fig. 2c and 2d**). Palmitate-induced TUNEL-positive apoptotic cell death also suppressed by rhVaspin (**Fig. 2e**). In addition, rhVaspin suppressed palmitate-induced accumulation of p62 with statistical significance (**Fig. 2f**)."

By the treatment with ER stress inducers (tunicamycin and thapsigargin), the average size of lysosomes increased by the analysis with lysotracker (**Supplementary Fig. 2a and 2c**). It is reported that mature Lamp1 is localized on lysosome membrane, while precursor molecules distribute through the rough ER to Golgi (*Arch Biochem Biophys.* 249(2):522-32, 1986). Tunicamycin inhibits oligosaccharide trimming and processing modifications of Lamp1 (*Arch Biochem Biophys.* 249(2):522-32, 1986), therefore the quantification of Lamp1 is not suitable for the measurement of lysosome size. The average size of Lamp1 particles was increased by thapsigargin but unaltered by tunicamycin (**Figures for reviewers, Fig. 4**), suggesting the differential effects of these agents on the maturation process of Lamp1 glycoprotein. Therefore, we evaluated lysosome size by lysotracker (**Supplementary Fig. 2a and 2c**), and impaired lysosome function by p62 accumulation (**Supplementary Fig. 2b and 2d**).

We are not able to quantify cathepsin B in IF images, because cathepsin B imaging was to demonstrate changes of cathepsin B distribution, *i.e.* punctuation pattern or diffuse distribution pattern in cytoplasm (**Fig 3d**). For the quantification, we demonstrate the levels of cathepsin B in cytosol fraction by Western blotting (**Fig 3e**).

5. In figure 4 the authors DO NOT demonstrate that IL-1 β increase or apoptosis is specific of proximal tubules, there is no staining clarifying this, so this claim is not supported by the data.

We thank reviewer #1 for the critical comment. We removed immunoperoxidase staining in **Fig. 4d** and the following description, "*IL-1 β expression in kidney tissues was increased in vaspin^{-/-} mice fed with HFHS diet, while it was ameliorated in vaspin Tg mice (Fig. 4d)*", from the manuscript.

6. *Figure 4: Could the authors comment on why there is no statistical significance in Figure 4 between the WT (HFHS) and Vaspin Tg (HFHS)? Does this mean that Vaspin does not protect against apoptosis? Why?*

We greatly appreciate reviewer #1 for the comment. This is a statistical test for 6 groups comparison by one-way ANOVA followed by Tukey–Kramer method, and p is lower than usual cutoff of $p < 0.05$. Since the transgenic and knockout mice experiments are distinct animal models, we now perform the statistical analysis in 4 groups; Vaspin Tg (HFHS), WT (HFHS), Vaspin Tg (STD), WT (STD), there is statistical difference between WT (HFHS) and Vaspin Tg (HFHS) ($p = 0.015$) by one-way ANOVA followed by Tukey–Kramer method. Similarly, we analyze 4 groups of Vaspin^{-/-} (HFHS), WT (HFHS), Vaspin^{-/-} (STD), WT (STD) in **Fig 4f**.

7. *Figure 6 and following: the authors say that they performed Anti-FLAG IP purification of transfected vaspin and then purified this band to depict interactors with LC-MS. From here they found the interaction with HSPA1L. The authors do not show the Proteomics data and data analysis, which should be shown at least in a supplementary figure, table etc.*

We completely agree with reviewer #1. We add data excel file as **Supplementary Table 2**.

8. *The critical interaction between vaspin and HSPA1L should be shown in a native system: It is one thing to show the interaction of a protein in an overexpression system and another to show that this interaction is physiologically relevant. If the proposed mechanism through which vaspin protects and ameliorates tubular damage is by binding to HSPA1L and causing further internalization (e.g. preventing secretion of HSPA1L), then the authors must show this interaction in the rhVaspin treated mice, or the Vaspin Tgs.*

We thank reviewer #1 for the constructive comment. We injected rhVaspin into mice, performed the immunoprecipitation study, and detected an interaction between vaspin and HSPA1L in a native system. We add an image in **Supplementary Fig. 6a** and describe in page 10 lines 3-6, "*The interaction of vaspin and HSPA1L was also confirmed in vivo. By using*

vaspin antibody, we immunoprecipitated the kidney lysates derived from the mice injected with rhVaspin, then we subjected them to SDS-PAGE and the detection with HSPA1L antibody (Supplementary Fig. 6a)."

9. *Figure 6f and g: the quantification is based on GAPDH expression but the representative blot has anti-tubulin as a loading control. This is an example of the general disorganization and inconsistencies found throughout the manuscript and figures.*

We thank reviewer #1 for the critical comment. We repeated Western blot and reanalyzed using anti- α -tubulin as a loading control in **Fig 6f and 6g**. Only for the Western blot analyses using cytosolic fractions, GAPDH was used as internal control in **Fig. 3b and 3e**.

10. *Also, in this experiment can the authors clarify if they add the BSA in the medium that is already supplemented with FBS or not? If it is, adding 10 mg/ml of BSA is equivalent to a 4+ proteinuria, which is massive proteinuria. If in addition to that the culture media has FBS too, I would be worried about the possibility of an osmotic stress that is not equivalent to the mannitol control.*

We apologize for our inaccurate description. We supplemented 1% FBS in medium with or without BSA. We revise the description in method section, page 20 lines 3-4, *"To examine an effect of overdose albumin on culture cells, Bovine Serum Albumin (BSA) was supplemented into culture medium, in which FBS was reduced to 1%."*

We also performed the experiments using 40 mM mannitol control, and we did not find any effects on HSPA1L protein levels. We add **Supplementary Fig. 6c** and describe in page 10 lines 17-18, *"Mannitol was used as osmotic control of BSA, and it did not alter the expression of HSPA1L (Supplementary Fig. 6c)."*

11. *Another inconsistent result: why does vaspin not prevent the secretion (in supernatant) of the overexpressed 3xFLAG-GFP-HSPA1L protein?*

We thank reviewer #1 for the important comment. We repeated the overexpression experiments of 3xFLAG-GFP-HSPA1L protein in both KEH293T and HK2 cells. The supernatants and whole cell lysates are clearly demonstrated in the title of each blot in **Fig. 7**. In **Fig. 7c and 7e**, Vaspin prevented the secretion of HSPA1L into culture medium. We describe in page 11 lines 15-19, *"In addition, rhVaspin administration suppressed BSA-induced secretion of HSPA1L along with reversing cellular HSPA1L levels (Fig. 7c and 7d). We again*

confirmed that BSA-induced exocytosis of HSPA1L into culture medium was inhibited by rhVaspin (Fig. 7e), and concurrently HSPA1L was restored in cytoplasm by rhVaspin in HK2 cells (Fig. 7g)."

12. *The finding that HSPA1L is found mainly in distal tubules of patients with proteinuria is not clear. The stainings should be improved. In addition, the fact that HSPA1L is present in the distal tubules rises a possibility of it being secreted by the distal tubule and not secreted to the distal tubule, it is not clear what is cause and effect. It is known that this protein is secreted by the collecting duct in response to environmental perturbations (see Pockley AG (2003) Heat shock proteins as regulators of the immune response. Lancet. 362:469–476). If the authors want to show definitive secretion, the presence of HSPA1L in the urine of mice should be evaluated, as well as the increase in urine in response to increased proteinuria. This finding should then be abrogated either in rhVaspin treated mice or in Vaspin Tg mice.*

We thank reviewer #1 for the critical comment for the localization and secretion of HSPA1L in the patients. We also appreciate for notifying us the important reference. We should add the staining with Tamm-Horsfall protein, distal tubule marker, but unfortunately the clinical samples of kidney tissues were limited, and we are not able to repeat the immunostaining. We also tried to detect urinary HSPA1L protein by western blotting analysis. However, we could not detect the HSPA1L protein in urine samples. In further experiment, we need to establish the HSPA1L ELISA system to detect the urinary HSPA1L protein. In a revised version of manuscript, we add arrows to indicate the distal tubules for the clarity in **Supplementary Fig. 11**. In addition, we describe the limitation of the study in page 17 lines 7-10, *"Further investigations are needed to explore the roles of Hsp 70s family in the kidney. Specifically, we are planning to measure of human urinary HSPA1L in patients with acute or chronic kidney diseases including DKD and demonstrate a clinical significance of HSPA1L."*

13. *The authors mention in the main text and in the figure legends that they studied these mice at 30 weeks of age. However, the methods say they were sacrificed at 24-weeks. Did the authors study mice after this stage and fully characterize a CKD phenotype?*

We apologize for the errors in methods. All mice were fed with high-fat high-sucrose or standard diet from 8 weeks of age to 30 weeks of age. We corrected 24-weeks to 30-weeks in method section in page 19 line 10.

14. *Please specify the time the cell cultures were incubated with rhVaspin.*

We thank reviewer #1 to improve the description of the culture conditions. In Figure legends, we add the incubation time as follow; in **Fig. 2**, “Recombinant human vaspin (rhVaspin) was administrated at 100 ng/ml with PA or CTRL for 24hr.”; in **Fig. 3**, “rhVaspin was administrated at 100 ng/ml for 24 hr.”; in **Fig. 4**, “rhVaspin was co-administrated at 100 ng/ml for 24 hr.”; in **Fig. 5d**, “Western blot analyses of HK2 cells cultured with 500 μ M PA, 1 μ g/ml TM and 1 μ M TG with or without 100 ng/ml rhVaspin for 24 hr.”; in **Fig. 6**, “**h.** HK2 cells overexpressing pGFP or pGFP-HSPA1L were cultured with 10 mg/ml BSA or 100 ng/ml rhVaspin for 24 hr. Western blot analyses demonstrated that HSPA1L ameliorated BSA-induced p62 accumulation. **i.** HK2 cells were incubated with 10 mg/ml BSA and/or 100 ng/ml rhVaspin for 24 hr. Vaspin inhibited BSA-induced p62 accumulation. **j.** Hepatic cell line, H4-II-E-C3 cells, were incubated with 10 mg/ml BSA and/or 100 ng/ml rhVaspin for 24 hr.”; in **Fig. 7c and 7d**, “HEK293T cells expressing p3xFLAG-GFP-HSPA1L were cultured with BSA at a dose of 0 or 10 mg/ml for 24 hr.”; in **Supplementary Fig. 2**, “Recombinant human vaspin (rhVaspin) was co-administrated at 100 ng/ml for 12 hr.”; in **Supplementary Fig. 3**, “rhVaspin was co-administrated at 100 ng/ml for 12 hr.”; in **Supplementary Fig. 4**, “rhVaspin was co-administrated at 100 ng/ml for 12 hr.”; in **Supplementary Fig. 5**, “Incubation time of TM, TG with rhVaspin was 24 hr.”; in **Supplementary Fig. 8**, “HK2 cells expressing p3xFLAG-GFP-HSPA1L or p3xFLAG-GRP78 were cultured with rhVaspin for 24 hr.”

15. Please provide quantification data for all Western blots. Conclusions should not be drawn from 1 experiment.

We thank reviewer #1 for the important comment. We performed quantifications of Western blots with Image J and add graphs to Figures (**Fig. 2d, 3c, 3f, 4b-d, 5d, 6g, and 7a-f**) and Supplementary Figures (**Supplementary Fig. 2d, 5, 6c, 6e, and 8a-d**).

16. Please add the appropriate kDa reference to all western blots and not only in some of them.

We apologize for the missing in kDa references. We add kDa reference to all Western blots.

17. In Figure 3b, there is clearly more protein loaded on the lane of the PA-treated cells, as evidenced by increased GAPDH. Therefore, the representative Western blot is not truly representative. Please repeat the experiment and show the resulting representative blot.

We apologize for the inequality for protein loading in SDS-PAGE. We repeated the

experiment and changed to a representative blot image in **Fig. 3b**.

18. *In Figure 3e, the representative Western blot also needs improvement.*

We apologize for the insufficient quality of Western blot. We repeated the experiment and changed images in **Fig. 3e**.

19. *Figure 4e, please show TUNEL staining of control tissues too.*

According to the comment from reviewer #1, we add figures of TUNEL staining of control tissues in **Fig. 4e**.

20. *Could the authors specify the TUNEL kit assay that was used?*

We apologize for the lack of some of method information. We added details of TUNEL assay in methods section in page 20 lines 5-8, "*TdT-mediated dUTP nick end-labeling (TUNEL) assay. MEBSTAIN Apoptosis TUNEL Kit III (Medical & Biological Laboratories, Nagoya, Japan) was used for TUNEL assay in culture cells. DeadEnd™ Colorimetric TUNEL System (Promega, WI, USA) was used for TUNEL assay in mouse kidney tissues.*"

21. *Quantification for Supplementary figure 5 would be desirable.*

According to the comment from reviewer #1, we performed the quantification of Western blot analysis and add graphs in **Supplementary Fig. 5**. For statistical analysis, we perform the multiple comparisons of the corresponding sets of negative control and experimental groups, as follows.

1. DMSO, TM, DMSO+rhVasipn, TM+rhVasipn in shRNA-CTRL
2. DMSO, TM, DMSO+rhVasipn, TM+rhVasipn in shRNA-GRP78
3. DMSO, TG, DMSO+rhVasipn, TG+rhVasipn in shRNA-CTRL
4. DMSO, TG, DMSO+rhVasipn, TG+rhVasipn in shRNA-GRP78

21. *The arrangement of Figure 6d is difficult to read. Please rearrange this figure, for example, using + or – signs.*

We thank reviewer #1 for the suggestion. We rearranged **Fig. 6d** by using + or – signs.

22. *Figure 6e needs to be repeated and add the same amount of protein in each lane, otherwise the expression levels are not comparable.*

We thank reviewer #1 for the comment to improve the quality of experiments. Although we loaded each 30 µg protein on gel, the expressions of tubulin were not equal in contrast to GAPDH. Li *et al.* reported that housekeeping proteins are used as internal controls in the Western blot analysis with presumed stability and no changes in physiological condition. However, if these housekeeping proteins DO change in certain biological or pathological condition, then using these housekeeping proteins as internal controls may cause problems in data acquisition, analysis, and interpretation (Life Sci. 92(13): 747–751, 2013). We repeated the experiments in **Fig 6e.** and **Figures for reviewers, Fig. 5,** showing 3 different housekeeping proteins and Ponceau-S staining. Ponceau-S staining shows that almost same amount of proteins in each sample was loaded.

23. Data analysis and Statistics: For studies shown in Figure 2, it looks like the comparisons have been designed to set the controls to one, and eliminate the variance of the control. This design makes statistical inferences prone to error. The data should be reanalyzed as in the other studies so that the variance of test and control groups is accounted for.

We appreciate reviewer #1 for the important comment about the statistical analyses. We reanalyzed the data and change the graph with statistical variance in control group in **Fig. 2.**

Reviewer #2 (Remarks to the Author):

This study had identified a novel role for the visceral adipose tissue derived serine protease inhibitor (Vaspin), an important adipokine, in the pathogenesis of diabetic kidney disease (DKD) through ameliorating ER stress, autophagy impairment and lysosome dysfunction in proximal tubular cells (PTCs). Using Vaspin^{-/-} KO mice in a high fat high sucrose diet (HFHS) model, they found prominently enlarged lysosomes in PTCs which was associated with increased cell apoptosis compared to that seen in their wild type litter mates. Furthermore, the authors also conducted mechanistic experiments in HK2 and HEK293T cell lines to test the role of vaspin in protecting PTCs from metabolic stress. Their findings indicate that intracellular vaspin levels are regulated by several molecular mechanism. Firstly, they found that heat shock protein family A (Hsp70) member 1 like (HSPA1L) as a new vaspin interactive protein in PTCs. Secondly, they found that GRP78, a cell-surface receptor, forms a complex with HSPA1L and clathrin heavy chain (CHC) respectively and are involved in endocytosis of vaspin. They conclude that vaspin may play a role as an effector of GRP78/HSPA1L signaling. Overall, this manuscript includes new and interesting data showing unique cross talk between adipose derived adipokine and PTC function

related to DKD. Vaspin appears to be a novel protective factor for proximal tubular cells in diabetes and obesity related DKD. The conclusions are derived from in vivo models, in vitro mechanistic data as well as some human relevant information. There are however several aspects to be addressed to strengthen the data and the manuscript.

Major points

1. *The authors should provide some results about the phenotype of the mice. Please show if there are any difference in body weight, blood glucose level, ITT, GTT, and adipocytes hypertrophy between Vaspin Transgenic, Vaspin^{-/-} KO and wild type mice.*

We appreciate reviewer #2 for the comment of the phenotypes in the experimental animals. We already reported the metabolic phenotypes of Vaspin Transgenic, Vaspin^{-/-} KO and wild type mice in previous publications (*Diabetes* **61**, 2823-2832, 2012; *Circ Res* **112**, 771-780, 2013). Therefore, in the present report, we did not include previously published data.

2. *Kidney histology shows severe tubular damage in Vaspin^{-/-} KO mice therefore it would be more interesting to show kidney dysfunction i.e urine Albumin/creatinine in these mice*

We appreciate reviewer #2 for the constructive comments. We measured the urinary albumin excretion and add the data in **Supplementary Fig. 1h** and describe in Result section, page 6 lines 9-12, "Urinary albumin tended to increase in the mice fed with HFHS diet compared with STD, although there were no statistical differences. Under HFHS diet, there were no differences of urinary albumin among Vaspin Tg, WT and Vapsin^{-/-} mice (**Supplementary Fig. 1h**)." We discuss about the lack of significant changes of albuminuria in response to reviewer #1 in page 3 lines 1-16.

3. *As Vaspin is an important adipokine to reduce ER stress induced by obesity, please explain and provide some results whether Vaspin^{-/-} KO STD mice developed any histopathological changes in adipose tissue and kidney compare to wild type mice.*

We thank reviewer #2 for the comment about the histological changes in adipose tissues. In previous studies, we reported the histopathological phenotype of adipose tissues in Vaspin^{-/-} STD mice and wild type mice (*Diabetes* **61**, 2823-2832, 2012). There were no histological differences in adipose tissues of wild and Vaspin^{-/-} mice fed with STD, although the adipocyte size of Vaspin^{-/-} mice was larger than wild type mice under HFHS diet. Similarly, there were no histological differences in kidney histology between Vaspin^{-/-} and

wild type mice under STD shown in **Fig. 1** and **Supplementary Fig. 1**.

4. *Western blot image in figure 2 does not show significant increase of GRP78 in PA treated cells - a quantitative validation is needed.*

We thank reviewer #2 for the comment for improvement of the study. We performed the quantification of Western blot and replace the representative images in **Fig. 2c and 2d**.

5. *Further experiments, i.e gene expression and IHC staining, should be performed to support the presence of HSPA1L in the mice kidney. Figure 6e is not solid evidence as the band is almost non-detectable.*

We thank reviewer #2 for the important comment which strengthens our manuscript. We purified total RNA from C57BL/6J mouse kidney tissue, and cDNA was synthesized. Then, HSPA1L gene expression was confirmed by PCR. We add the agarose gel image of PCR in **Supplementary Fig. 6b** and describe in page 10 lines 8-11, "*Recently, Wang et al. reported mRNA expressions of Hspa1l in mouse heart, liver, spleen, ling, kidney, brain, muscle, and testis*²⁹. We also detected the gene expression of Hspa1l in C57BL/6J mouse kidney tissues (**Supplementary Fig. 6b**)."

We also repeated Western blot analysis in **Fig. 6e** and performed IHC staining of HSPA1L and add the images of kidney tissues in **Supplementary Fig. 6d**.

6. *What is the effect of ER stress on HSPA1L?*

We thank reviewer #2 for asking the effect of ER stress on the expression of HSPA1L. Although the excess amount of BSA reduced the expression of HSPA1L in HK2 cells, tunicamycin and palmitic acid did not alter the expression of HSPA1L in HK2 cells. We performed the quantification of HSPA1L in HK2 cells and add graphs in **Fig. 6g**.

7. *Vaspin Transgenic and Vaspin^{-/-} KO mice have been used in previous published paper by authors. However, for the current study a clear description of mice model should be provided.*

According to the suggestion from reviewer #2, we add a description of phenotypes of Vaspin Tg and KO mice in page 3 line 23 to page 4 line 3; "*In brief, body weight gain, fatty liver and insulin resistance were improved in transgenic (Tg) mice under high fat-high sucrose (HFHS) diet, while knockout (KO) mice revealed deterioration of hepatosteatosis and insulin resistance*^{11, 12}. *Vaspin gene transfer ameliorated balloon-injured intimal proliferation in WKY rats and it protected*

from apoptosis of cultured human aortic endothelial cells^{11, 12}. Vaspin contributes to favorable roles on hepatocyte or endothelial cell via binding to cell surface 78kDa glucose-regulated protein (GRP78)^{11, 12}”.

8. Please mention HFHS diet components. How much calories come from fat /carbohydrate /sugar?

We thank reviewer #2 for the comment to improve the method description. We add the information of commercially available chow diet in method section page 19 lines 7-9, “Eight-week-old mice were subjected to standard diet (STD) group or high fat-high sucrose diet (HFHS) (5.56 kcal/g; fat 58 %, carbohydrate 25.5%, and protein 16.4 %) (D12331: Research Diet) group.”

9. In the methods, authors have mentioned that 24-week-old kidneys were collected, but results indicate kidneys were from 30 week old mice? Please specify the exact number of weeks that the mice were under HFHS diet and when they were euthanized to collect kidney samples.

We apologize for the errors for the description in the age of mice. All mice were fed with high-fat high-sucrose or standard diet from 8 weeks of age to 30 weeks of age. We correct 24-weeks to 30-weeks in method section in page 19 line 10.

10. In the cell culture studies please mention the concentration of low and high glucose treatments.

We thank reviewer #2 for a comment. We add the glucose concentrations in culture medium in page 19 lines 15-19, “HK2 cells were cultured in DMEM with low glucose (5.5 mM glucose), GlutaMAX™ Supplement and pyruvate (GIBCO), and supplemented with 10% fetal bovine serum (FBS), 100 U/ml penicillin and 100 µg/ml streptomycin at 37°C in 5% CO₂. HEK293T cell were cultured in DMEM with high glucose (25 mM glucose), GlutaMAX™ Supplement (GIBCO), and supplemented with 10% FBS, 100 U/ml penicillin and 100 µg/ml streptomycin.”

11. Please provide the catalog number and concentrations of all primary and secondary antibodies used for IF,IHC, and western blot experiments

We thank reviewer #2 for a suggestion. We add the description about the catalog numbers

and concentrations of all antibodies in pages 21-24.

Here are some recommendations for the Figures:

Figure 1: more details and explanation of the results seen needs to be added to the figure legend.

- Show vacuoles and Lamp1 positive lysosomes using arrows.
- Mention that different magnification has been used for images g-i and j-l)

We thank reviewer #2 for the recommendations to figures. We add the description of the results in **Fig. 1** legends, “In *Vaspin*^{-/-} mice fed with HFHS diet, prominent vacuolations are observed in renal tubular cells (c), while they are ameliorated in *Vaspin* Tg mice (a). Vacuolations are shown by arrows. Immunohistological staining of lysosomal-associated membrane protein 1 (*Lamp1*) (g-o). These vacuoles are positive for lysosomal-associated membrane protein 1 (*Lamp1*). *Lamp1*-positive lysosomes are shown by arrow heads. In *Vaspin*^{-/-} mice under HFHS diet, the number of large lysosomes is increased (i and l) compared with WT mice (h and k). HFHS diet-induced lysosomal enlargement is inhibited in *Vaspin* Tg mice (g and j).”

As suggested by reviewer #2, we show vacuoles and Lamp1 positive lysosomes using arrows in **Fig. 1g-1l**. We also describe that “Different magnifications are used for images g-i and j-l, Bar=100 μm”, in the figure legend.

Figure 2:

- Please add markers for western blot images
- Please provide more specific details about statistical analysis if one way ANOVA followed by post hoc tests to explore differences between multiple groups.
- P values are missing from the legends

We thank reviewer #2 for the suggestions to **Fig. 2**. We add the marker for Western blot images. We apologize for the missing description of statistical analyses in method section. In revised version manuscript, we add the description of details about statistical analysis in page 25. In addition, we add p values in the legends.

Figure 3:

- Please follow chronological order when writing legends for this figure i.e. a-d.

We thank reviewer #2 for the suggestions to **Fig. 3**. We revise the figure legends in chronological order.

Figure 4:

- Western blots – please add Molecular weight markers to indicate which band has been quantified.
- Please add the names of NLRP3 inflammasome pathway proteins in the legend.

We thank reviewer #2 for the suggestions to **Fig. 4**. We add the molecular weight markers. In the legends, we add the names of NLRP3 inflammasome pathway as follow, “*Western blot analysis of NLRP3 inflammasome pathway related proteins; NLRP3, Caspase 1 and IL-1 β .*”

Figure 6:

- Western blot pictures need marker to detect bands.

We thank reviewer #2 for the suggestions to **Fig. 6**. We add the molecular weight markers.

Figure 1 supplementary:

- Please follow chronological order when writing legends for this figure i.e. a-d.
- Indicate lysosomes using arrow in electron micrographs.

We thank reviewer #2 for the suggestions to **Supplementary Fig. 1**. We revise the legends in chronological order, and indicate the lysosomes using arrows in EM.

Fig 2 supplementary:

- Western blot pictures need molecular weight markers.
- Please say the white arrows indicate positive apoptotic cells in the legends.
- P values are missing in the figure legends.

We thank reviewer #2 for the suggestions to **Supplementary Fig. 2**. We add the molecular weight markers in Western blot images, an explanation of white arrows showing apoptotic cells, and *p* values in the figure legends.

Figure 8 supplementary:

- The legend is not very clear and the figure needs more clear labeling.

We thank reviewer #2 for the suggestions to **Supplementary Fig. 8**. We label the fraction numbers on the images and revise the legends with clear description. The figure number is

changed to **Supplementary Fig. 9**.

Figure 9 supplementary:

- Please provide key to the abbreviation on the figures to improve clarity
- Include the complete name for U-pro in the legend.
- Rearrange the images based on the order of results.
- Mention figure number in the Results text accordingly.

We thank reviewer #2 for the suggestions to **Supplementary Fig. 9**. We add the complete abbreviations on the bottom of figure such as “DN; diabetic nephropathy, TBMD; thin basement membrane disease, ORKD; obesity-related kidney disease”. We also describe the abbreviations in the legends. We also rearrange the images based on the order of results and changed figure number in the results text. The figure number is changed to **Supplementary Fig. 10**.

Reviewers' comments:

Reviewer #1 (Remarks to the Author):

In this revised manuscript, Nakatsuka et al. have generally been responsive to the comments on the previous version.

The central premise, that vaspin partially reverses the effects of palmitic acid (and other ER stressors) is supported by the data but there are so many tangents to this central message contained in this manuscript, that it becomes confusing. The manuscript would improve by a focus on better writing and organization, and also a rationale/ justification as to why the authors have chosen to move from one cellular model to another (HEK, HK2 etc), which is not addressed. Also, the animal models are not fully exploited to support all the mechanistic findings presented (e.g. p62 staining in vivo, please see below).

A conceptual question that remains confusing: why is it that rhVaspin prevents p62 increase but not LC3?

Specific comments:

Fig. 5C

- HSPA1L is shown here but the authors haven't mentioned the interaction with this protein as yet in the manuscript. Notably, the Western blot shows that HSPA1L is reduced with TG treatment, but there is no comment about this effect in the text/legend.

Fig.6

- This Figure should be rearranged, it is very difficult to understand as presented.
- Panel h. Could the authors comment on why is p62 increased with BSA treatment? This should not be the case...

Of note, in Supplementary Fig. 6, the IP is of poor quality such that the IP band is difficult to see, so this IP quality should be improved.

Fig. 7

Please note that Line 295 refers to Fig. 7g but it should be Fig. 7f.

Supp Fig. 8 and Supp Fig. 9

Line 301-303 "In addition, co-administrations of anti-GRP78 301 antibody or anti-HSPA1L antibody with rhVaspin in HK2 cells, vaspin protein levels of cell lysate were decreased (Supplementary Fig.8c). It is hard to understand what is meant here, and, if this finding is as important as it seems, then it would be a good idea to move Supp Fig 8g, Supp Fig 9 and 13 to the main Figures/text.

Discussion

A schematic representation of the proposed mechanism to aid the reader will be very helpful.

Line 362: "PTCs revealed autophagy impairment demonstrated by p62 accumulation." This is not precise, the authors show p62 accumulation in HK2 cells, which is an immortalized in vitro cell culture system. There is no histologic (or IF) evidence to show that in vivo PTCs of vaspin-/- mice fed HFHS diet have p62 accumulation (which is an experiment that can be easily done).

Line 436 "HSPA1L was strongly expressed in distal tubules rather than proximal tubules. One hypothesis is that secreted HSPA1L from PTCs reached to distal tubular cells. Jheng. HF et...". Since it is not shown in vivo, that HSPA1L is secreted, it is very speculative to claim that the reason there is increased expression in the distal tubule is secretion (one would also have to invoke that the protein is taken up by the distal tubular cells, which the authors have no evidence for).

Reviewer #2 (Remarks to the Author):

The authors have addressed all my comments in a satisfactory manner

Responses to reviewer #1

The central premise, that vaspin partially reverses the effects of palmitic acid (and other ER stressors) is supported by the data but there are so many tangents to this central message contained in this manuscript, that it becomes confusing. The manuscript would improve by a focus on better writing and organization, and also a rationale/ justification as to why the authors have chosen to move from one cellular model to another (HEK, HK2 etc), which is not addressed. Also, the animal models are not fully exploited to support all the mechanistic findings presented (e.g. p62 staining in vivo, please see below).

Why the authors have chosen to move from one cellular model to another (HEK, HK2 etc), which is not addressed.

We greatly appreciate for the constructive comment from reviewer #1. We described the justification and rationale for using HEK293T and HK2 cells in lines 228-230, *“To demonstrate the protein interactions of vaspin with GRP78 and HSPA1L, HEK293T cells were employed because of better scale-up efficiency for protein preparation in immunoprecipitation studies.”* and in lines 230-232, *“To demonstrate the specific cellular responses of PTCs to various stress, we used HK2 cells in following experiments.”*

A conceptual question that remains confusing: why is it that rhVaspin prevents p62 increase but not LC3?

We thank reviewer #1 for the important comment. We explain the conceptual interpretation in result section in lines 163-168, *“The p62 protein, which is ubiquitin-binding scaffold protein, and LC3 cooperates to deliver the waste materials to autophagosomes for degradation. In general, an increase in both LC3-II and p62 parallels in the enhancement of autophagy or lysosomal dysfunction. Since we observed that rhVaspin reversed p62 but not LC3, we further moved to investigate the effects of rhVaspin on established ER stress inducers.”*

Specific comments:

Fig. 5C

- HSPA1L is shown here but the authors haven't mentioned the interaction with this protein as yet in the manuscript. Notably, the Western blot shows that HSPA1L is reduced with TG treatment, but there is no comment about this effect in the text/legend.

We thank reviewer #1 for an important comment. We carefully repeated Western blot analyses of GRP78, HSPA1L and CHC using cell surface protein with Ponceau-S staining.

New gel images with triplicate or quadruplicate samples demonstrates cell surface expressions of HSPA1L was mildly increased by tunicamycin (TM) or thapsigargin (TG) administration. We replaced the Figure 5c and revised results in lines 313-315, "*HSPA1L was detected in cell surface proteins as the case with GRP78 (Fig. 5c). However, the increase in cell surface HSPA1L by TM and TG was mild and the treatment with palmitate rather decreased cell surface HSPA1L (Fig. 5c).*" and Figure legends in lines 861-863, "*Cell surface GRP78 is prominently increased by PA, TM and TG. In contrast, HSPA1L is mildly increased by TM and TG, but rather decreased by PA.*"

Fig.6

- This Figure should be rearranged, it is very difficult to understand as presented.
- Panel h. Could the authors comment on why is p62 increased with BSA treatment? This should not be the case...

Of note, in Supplementary Fig. 6, the IP is of poor quality such that the IP band is difficult to see, so this IP quality should be improved.

We thank reviewer #1 for the constructive comments. To assist the understanding of readers, we rearranged Fig. 6b and 6c by the combination of presence (+) and absence (-) of plasmids and recombinant protein.

In panel h, p62 increased by the treatment with 10 mg/ml BSA and the accumulation of p62 was ameliorated by HSPA1L overexpression. Liu *et al.* reported that urinary protein triggered defective lysosomal acidification and LMP (Am J Physiol Renal Physiol 308: F639–F649, 2015). The alteration of intralysosomal pH decreases lysosomal enzymatic activity and degradation activity, and subsequently promotes autophagy impairment indicated by p62 accumulation. Yamamoto *et al.* reported that 2.5 mg/ml BSA treatment increased the degradation of p62 protein in a time-dependent manner, as shown by the p62 turnover index (J Am Soc Nephrol 28: 1534–1551, 2017). In Supplemental Figure 3B published in JASN article, BSA treatment increased p62 protein levels compared to non-treated control cells. We briefly described the previously published data in lines 263-265, "*Since albuminuria is major driving force for the progression of renal dysfunction and BSA is known to induce the lysosomal dysfunction and accumulation of p62^{15,30}, we further applied BSA in HK2 cell culture.*"

In **Supplementary Fig. 6a lower panel**, we further performed the immunoprecipitation with anti-HSPA1L antibody using kidney samples of rhVaspin-injected mouse and they were blotted by anti-Vaspin antibody. In **Supplementary Fig. 6a upper panel**, we replaced the images of immunoprecipitation with anti-Vaspin antibody and blotted by HSPA1L antibody with longer exposure of the membranes of previous revision.

Fig. 7

Please note that Line 295 refers to Fig. 7g but it should be Fig. 7f.

We thank reviewer #1 for the comment. We corrected Fig.7g to Fig. 7f.

Supp Fig. 8 and Supp Fig. 9

Line 301-303 *"In addition, co-administrations of anti-GRP78 301 antibody or anti-HSPA1L antibody with rhVaspin in HK2 cells, vaspin protein levels of cell lysate were decreased (Supplementary Fig.8c). It is hard to understand what is meant here, and, if this finding is as important as it seems, then it would be a good idea to move Supp Fig 8g, Supp Fig 9 and 13 to the main Figures/text.*

We apologize for inaccurate description. Our aim of the experiment of co-administrations of anti-GRP78 or anti-HSPA1L antibody with rhVaspin in HK2 cells is to demonstrate blocking a binding between Vaspin and GRP78 / HSPA1L decrease endocytosis of vaspin and cellular vaspin protein level. We added detail explanation in the legends of **Supplementary Figure 8c**, *"HK2 lacks intrinsic expression of vaspin and the internalization of rhVaspin into HK2 cells is inhibited by both antibodies."* As reviewer #1 recommended, we moved **Supplementary Figures 8g, 9 and 13** to main **Figures 8-10**.

Discussion

A schematic representation of the proposed mechanism to aid the reader will be very helpful.

We thank reviewer #1 for the suggestion. We drew schema of proposed mechanism in **Supplementary Figure 12**, and described in the legends, *"Proposed mechanism of vaspin protecting proximal tubular cells. Vaspin protects proximal tubular cells (PTCs) coordinating with GRP78 and HSPA1L, through ameliorating excessive ER stress, autophagy failure, lysosomal membrane permeabilization, NLRP3 inflammasome activation and cell death. Albumin-induced depletion of HSPA1L of PTCs along with increased HSPA1L secretion is a novel mechanism of albuminuria-induced disturbance in homeostasis of PTCs."*

Line 362: *"PTCs revealed autophagy impairment demonstrated by p62 accumulation."* This is not precise, the authors show p62 accumulation in HK2 cells, which is an immortalized in vitro cell culture system. There is no histologic (or IF) evidence to show that in vivo PTCs of vaspin^{-/-} mice fed HFHS diet have p62 accumulation (which is an experiment that can be

easily done).

We greatly appreciate reviewer #1 for the comment, which strengthens our manuscript. We performed immunohistochemical staining of p62, and images were presented in **Supplementary Figure 1f and 1g**. We described in the legends, “*In immunohistochemical staining of tubules, HFHS diet-increased p62 accumulation was enhanced in Vapsin^{-/-} mice and it was ameliorated in Vapsin Tg mice (Supplementary Fig. 1f and 1g).*”

Line 436 “HSPA1L was strongly expressed in distal tubules rather than proximal tubules. One hypothesis is that secreted HSPA1L from PTCs reached to distal tubular cells. Jheng. HF et....”. Since it is not shown *in vivo*, that HSPA1L is secreted, it is very speculative to claim that the reason there is increased expression in the distal tubule is secretion (one would also have to invoke that the protein is taken up by the distal tubular cells, which the authors have no evidence for).

We thank reviewer #1 for the critical comments. We deleted a sentence of “*One hypothesis is that secreted HSPA1L from PTCs reached to distal tubular cells.*” And we described the limitations in lines 439-443, “*However, we failed to detect urinary HSPA1L excretion in mice by Western blot analysis, therefore one of the limitations of current study is that albumin-induced HSPA1L secretion is not evidenced by *in vivo* study. The development of high-sensitive ELISA system to detect HSPA1L would confirm this hypothesis.*”

REVIEWERS' COMMENTS:

Reviewer #1 (Remarks to the Author):

All my comments have been adequately addressed and I commend the authors for making the effort to address them.